# Benchmarking and Evolving
# Reason-Reflect-Rectify for Reflective Visual Generation

Junjie Wang [* 1]   Xinghua Lou [* 2 3]   Xiangtai Li [4]   Ye Tian [5]   Keyu Chen [1]   Yulin Li [1]   Bin Kang [6]   Guangcan Mai [7]
Yanwei Li [8]   Zhuotao Tian [1 3 †]   Liqiang Nie [1 3]

## Abstract

Text-to-Image (T2I) models and Unified Multimodal Models (UMMs) have achieved remarkable progress in visual generation. However, their reliance on a single-pass generation paradigm limits their ability to handle complex prompts requiring iterative refinement. To enable multi-round Reflective Visual Generation (RVG), we formalize the *Reason–Reflect–Rectify* (R³) loop as a core framework and introduce R³-Bench, a benchmark of over 600 expert-annotated instances that quantifies iterative reasoning and rectification capabilities. Evaluation on R³-Bench reveals a critical gap: while state-of-the-art models can identify generation errors, they fail to generate actionable rectification instructions. To bridge this gap, we propose R³-Refiner, a dual-stage framework leveraging Group Relative Policy Optimization (GRPO) and a Hierarchical Reward Mechanism (HRM) to better align rectification with reflective reasoning. Experiments show that R³-Refiner achieves significant improvements on R³-Bench (+12.0% in Reflective Verdict Score, +9.0% in Rectification Score), and can be seamlessly integrated with various MLLMs to enhance the generation quality of different T2I models on GenEval++ and T2I-CompBench. Code is available at https://github.com/xiaomoguhz/R3-Bench.

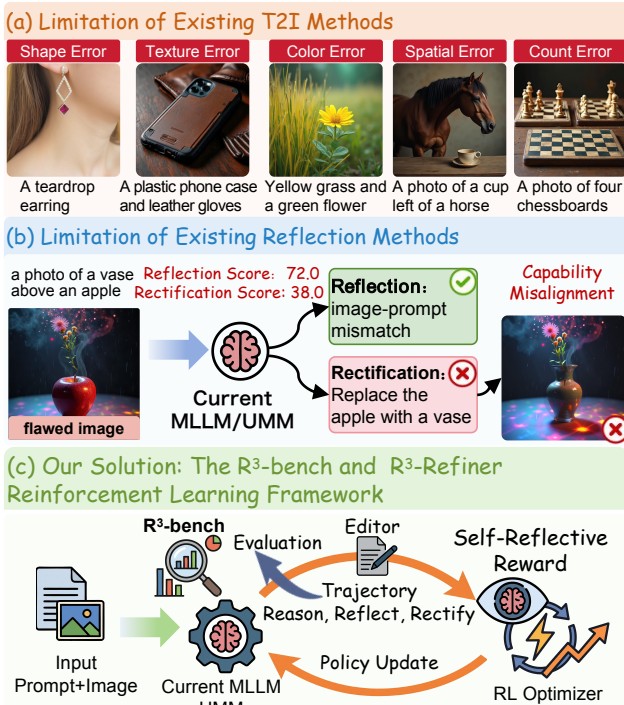

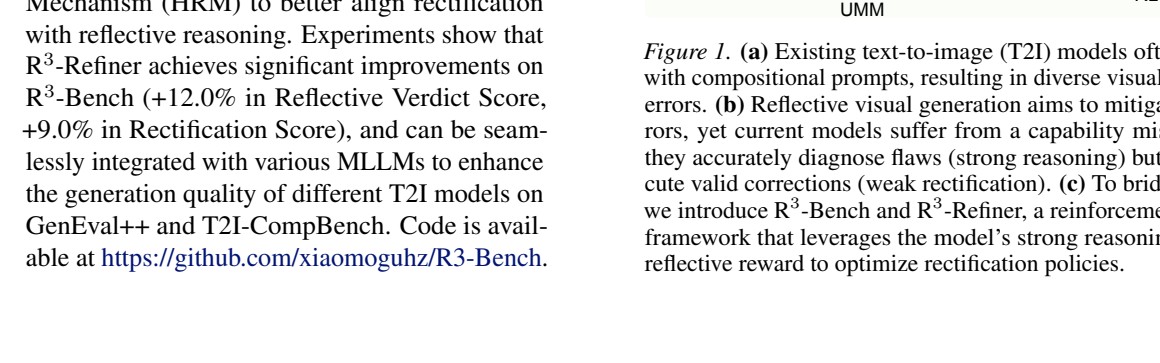

*Figure 1.* **(a)** Existing text-to-image (T2I) models often struggle with compositional prompts, resulting in diverse visual generation errors. **(b)** Reflective visual generation aims to mitigate these errors, yet current models suffer from a capability misalignment: they accurately diagnose flaws (strong reasoning) but fail to execute valid corrections (weak rectification). **(c)** To bridge this gap, we introduce R³-Bench and R³-Refiner, a reinforcement learning framework that leverages the model's strong reasoning as a self-reflective reward to optimize rectification policies.

---

[*]Equal contribution  [1]Harbin Institute of Technology, Shenzhen [2]University of Science and Technology of China [3]Shenzhen Loop Area Institute [4]Nanyang Technological University [5]Peking University [6]University of Chinese Academy of Sciences [7]Hong Kong Baptist University [8]Shanghai Jiao Tong University. Correspondence to: [†]Zhuotao Tian <tianzhuotao@gmail.com>.

*Proceedings of the 43$^{rd}$ International Conference on Machine Learning*, Seoul, South Korea. PMLR 306, 2026. Copyright 2026 by the author(s).

## 1. Introduction

Text-to-Image (T2I) task (Esser et al., 2024; Podell et al., 2023; Labs, 2024; Labs et al., 2025; Rombach et al., 2022) has achieved remarkable success with diffusion models. Building upon these advancements, Unified Multimodal Models (UMMs) (Xie et al., 2025a; Liao et al., 2025; Chen et al., 2025c; Xin et al., 2025; Yang et al., 2025d; Huang et al., 2025b; Cui et al., 2025) integrate the reasoning capabilities of Multimodal Large Language Models

(MLLMs) (Li et al., 2023; Team et al., 2025; Yang et al., 2025b; Guo et al., 2025; Lu et al., 2024; Liu et al., 2023; Zhu et al., 2023; Wang et al., 2025b) to further enhance visual generation capabilities. However, as illustrated in Fig. 1(a), these models still struggle with compositional prompts because the open-loop, single-pass generation paradigm lacks mechanisms for error rectification. To overcome this limitation, a transition to a multi-round Reflective Visual Generation (RVG) paradigm is necessary.

**Insufficient Evaluation for RVG.** Following the success of self-reflection mechanisms in Large Language Models (LLMs) (Yao et al., 2022; Shinn et al., 2023; Ma et al., 2025; Huang et al., 2025d; Chen et al., 2025b) and MLLMs (Ding & Zhang, 2025; Zhang et al., 2025a; Wang et al., 2025c; Kumar et al., 2024; Madaan et al., 2023), recent visual generation studies (Huang et al., 2025c; Zou et al., 2025; Gu et al., 2025) have started exploring closed-loop RVG paradigms. However, advancing research in this direction is hindered by a critical *evaluation* gap. As illustrated in Fig. 2(a), existing benchmarks predominantly measure isolated capabilities, including attribute alignment (Ye et al., 2025; Ghosh et al., 2023; Hu et al., 2024), reasoning and knowledge-based generation (Niu et al., 2025; Wu et al., 2025e), or multidimensional understanding and generation (Xie et al., 2025b; Chang et al., 2025; Shi et al., 2025). None of these benchmarks quantifies the iterative reasoning processes integral to RVG, *i.e.*, diagnosing visual inconsistencies, reflecting upon corrective strategies, and rectifying generated outputs.

**The Proposed $R^3$-Bench.** To effectively assess RVG capabilities, we formalize the critical competencies into the Reason-Reflect-Rectify ($R^3$) loop and introduce the $R^3$-Bench, which comprises 670 expert-annotated correction tasks covering both synthetic and real-world scenarios. Each task provides a textual prompt paired with a flawed generated image, requiring the model to output a structured response, consisting of a *verdict* answer, a *reflective* explanation, and a *rectification* action for evaluation. We employ a dual evaluation protocol to comprehensively assess the model (Fig. 2(b)). Specifically, we evaluate the diagnostic accuracy of the verdict and reflective explanation while quantifying the efficacy of the rectification action based on the relative visual improvement of the flawed image.

**Key Observations.** Using our benchmark, we find that even state-of-the-art visual reasoning and generation models fall short in these challenging scenarios. As shown in Fig. 1(b), although leading MLLMs (Bai et al., 2025b; Zhang et al., 2025b) can identify inconsistencies between textual prompts and generated images, they often fail to yield actionable rectification instructions. Such a discrepancy limits the effectiveness of the closed-loop RVG pipeline required for high-quality visual generation. This

observation raises a critical question: *Can we harness the model's strong discriminative capability as a reward signal to enhance its rectification ability through self-evolution?*

**Our Solution.** To address the issue, we propose $R^3$-*Refiner*, a reinforcement-learning-based refinement framework for RVG, as shown in Fig. 1(c). $R^3$-Refiner explicitly adopts the Reason–Reflect–Rectify ($R^3$) loop and aligns rectification behavior with reflective reasoning through structured self-reward. Given a misaligned text-image pair, $R^3$-Refiner first produces a structured $R^3$ trajectory, consisting of (i) a reasoning process that diagnoses visual inconsistencies, (ii) a reflective verdict that determines error types and correction necessity, and (iii) a rectification instruction specifying actionable edits.

$R^3$-Refiner adopts a *Hierarchical Reward Mechanism* (HRM) for optimization, which decomposes supervision across different stages of the $R^3$ loop, as illustrated in Fig. 4. Specifically, the reasoning and reflection stages are supervised using constructed $R^3$ data, ensuring accurate inconsistency diagnosis and reliable reflective judgments. For rectification, $R^3$-Refiner adopts a closed-loop verification process where the rectification instruction is executed by an external image editor to generate a revised image, which is subsequently re-evaluated by the model to derive a self-reward signal based on visual improvement. Experiments on Qwen2.5-VL and Qwen3-VL (Bai et al., 2025a) demonstrate that $R^3$-Refiner enhances both reflective verdict score and rectification score of the baseline. Moreover, $R^3$-Refiner can be seamlessly integrated with various MLLMs to enhance the generation quality of different T2I models, such as Bagel, OmniGen2, and GPT-Image.

In summary, the contributions of this work are as follows:

- We introduce $R^3$-Bench, a benchmark that operationalizes RVG through the Reason-Reflect-Rectify ($R^3$) loop and evaluates models across diagnosis, reflection, and rectification capabilities.

- Utilizing $R^3$-Bench, we identify a critical capability misalignment between discriminative reasoning and rectification execution. To address this, we propose $R^3$-Refiner, a dual-stage framework that optimizes the complete $R^3$ loop using Group Relative Policy Optimization (GRPO) and the HRM.

- Experiments demonstrate that $R^3$-Refiner significantly improves performance on $R^3$-Bench (+12.0% reflection, +9.0% rectification) and can be integrated with various MLLMs to enhance the generation quality of various T2I models.

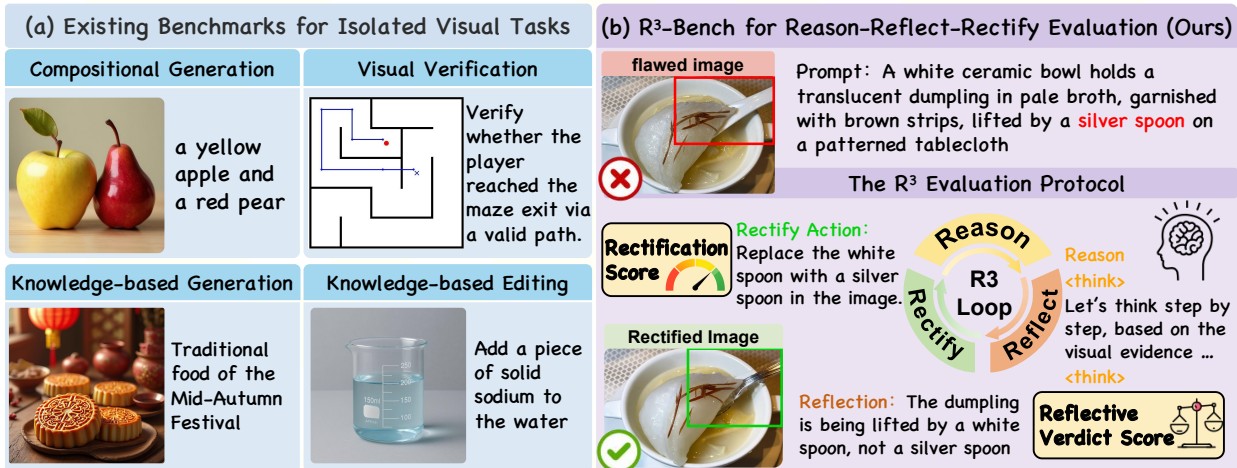

*Figure 2.* **Comparison between existing benchmarks and R³-Bench.** **(a)** Existing benchmarks predominantly evaluate image generation, editing, and visual verification as isolated tasks. **(b)** In contrast, R³-Bench centers on the "Reason-Reflect-Rectify" loop for Reflective Visual Generation (RVG). As illustrated by the "silver spoon" example, the model first employs reasoning to diagnose inconsistency by providing a verdict and a reflective explanation. The accuracy of this diagnosis is quantified by the Reflective Verdict Score. Subsequently, the model generates a rectification action to guide the refinement process. The efficacy of this correction is measured by the Rectification Score, which assesses relative visual improvement.

## 2. Reason-Reflect-Rectify for Reflective Visual Generation

This section introduces the task definition and benchmark construction for the Reason-Reflect-Rectify (R³) framework. We first formalize the iterative R³ loop tailored for RVG tasks in Sec. 2.1. Next, Sec. 2.2 describes the composition and construction pipeline of the benchmark. Finally, Sec. 2.3 outlines the corresponding evaluation protocol.

### 2.1. Task Formalization

We formalize RVG as an iterative Reason-Reflect-Rectify (R³) loop that progressively refines image generation outputs. Unlike the conventional single-pass generation paradigm, the R³ loop involves iterative processing by an MLLM or UMM, consisting of three stages: a global binary verification of image-text consistency (*reason*), detailed localization of semantic discrepancies (*reflect*), and formulation of precise corrective actions (*rectify*).

**The Iterative R³ Loop.** Formally, at refinement step $t$, the task aims to learn a policy $\pi_\theta$ that produces a structured response $R_t$ conditioned on the textual prompt $P$ and the current image $\mathbf{I}^{(t)}$. Adhering to the R³ framework, the structured response $R_t$ is a tuple comprising a verification answer $v_t$, a discrepancy explanation $e_t$, and a rectification action $a_t$:

$$R_t = \pi_\theta(P, \mathbf{I}^{(t)}) = \langle v_t, e_t, a_t \rangle, \quad (1)$$

where $v_t$ serves as a global binary indicator for image-text consistency, $e_t$ localizes and explains semantic discrepancies in detail, and $a_t$ specifies targeted editing instructions

necessary to address the identified inconsistencies. Then, the structured response $R_t$ is used for iterative refinement.

Specifically, at iteration $t$, if the verification answer $v_t$ indicates image-text misalignment (denoted as *False*), the explanation and rectification tuple $\langle e_t, a_t \rangle$ instructs an external generative editor $\Phi$. Consequently, the editor modifies $\mathbf{I}^{(t)}$ to yield an improved image $\mathbf{I}^{(t+1)}$:

$$\mathbf{I}^{(t+1)} = \Phi(\mathbf{I}^{(t)}, \langle e_t, a_t \rangle). \quad (2)$$

This iterative refinement continues until the image-text consistency is confirmed, with $v_t$ becoming *True*.

### 2.2. Benchmark for Evaluation

In this section, we introduce R³-Bench, which assesses a model's proficiency in verifying image-text consistency, localizing semantic discrepancies, and formulating precise rectification actions. Below, we describe the dataset composition and construction pipeline in detail.

**Benchmark Overview.** As illustrated in Fig. 3, R³-Bench comprises 670 expert-annotated image-text pairs designed to evaluate the capabilities required for the iterative R³ loop. R³-Bench contains diverse error categories and balances aligned and misaligned instances to ensure comprehensive coverage. All instances are annotated with Ground-Truth (GT) verification labels to assess the accuracy of $v_t$. For misaligned instances, R³-Bench provides additional GT explanations and related visual question-answering (VQA) tasks. These annotations are essential for benchmarking the full R³ loop, enabling the evaluation of discrepancy reasoning $e_t$ and the effectiveness of rectification actions $a_t$

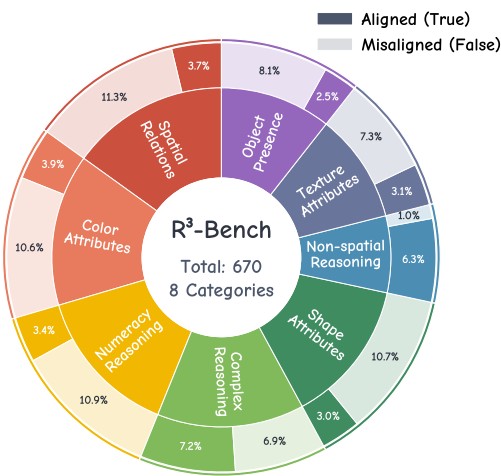

*Figure 3.* **Overview of R³-Bench.** The benchmark covers eight categories sourced from both real-world and model-generated data, comprising 222 aligned and 448 misaligned instances.

on rectified images.

**Benchmark Construction.** We construct the benchmark through a multi-stage process designed to ensure clarity and diversity. To form an initial candidate pool, we aggregate data from complementary sources, combining error samples generated by T2I models (Wu et al., 2025a) using prompts adapted from T2I-R1 (Jiang et al., 2025) and GenEval++ (Ye et al., 2025) with image–text mismatches obtained by rewriting samples from the GEdit dataset (Liu et al., 2025c) to incorporate diverse real-world domains.

This pool subsequently undergoes a cascaded filtering procedure (Sec. 3.2) to isolate preliminary matched and mismatched image–text pairs. Following this, we leverage MLLMs (Bai et al., 2025a) and LLMs (Yang et al., 2025a) to generate discrepancy explanations and VQA pairs that facilitate the evaluation of image improvements. To ensure the highest quality, human experts verify each case to refine annotations and eliminate ambiguous instances such as minor color distinctions or inconsistent scene atmospheres. This rigorous process yields 670 high-quality samples for balancing testing cost and data diversity. Examples and additional details are provided in Fig. 11 and Appendix G.

### 2.3. Evaluation Protocol

We evaluate models in the R³ loop using a two-phase protocol. Let $\mathcal{S} = \{(P_i, \mathbf{I}_i^{(t)}, v_i, e_i, Q_i)\}_{i=1}^N$ denote the test set, where $v_i$ and $e_i$ are the GT verification label and explanation, and $Q_i$ is a set of visual questions targeting key attributes in the prompt $P_i$.

**Phase I: Verdict-Reflection Alignment.** This phase evaluates the accuracy of the model's generated verdict $\hat{v}_i$ and reflection $\hat{e}_i$. To quantify their combined accuracy, we in-

troduce the Reflective Verdict Score ($\mathcal{S}_{\mathrm{ref}}$), as detailed in Appendix E.1. Specifically, we compute a correctness score $s_i \in \{0, 1\}$ for each sample based on the GT $v_i$. For aligned samples where $v_i = \text{True}$, the score depends solely on the predicted verdict and is defined as $s_i = \mathbb{I}(\hat{v}_i = \text{True})$. Conversely, for misaligned samples where $v_i = \text{False}$, we enforce a stricter criterion that requires correctness in both the verdict and the reflection. Accordingly, we define $s_i = \mathbb{I}(\hat{v}_i = \text{False}) \cdot \mathcal{J}(e_i, \hat{e}_i)$, where the LLM-Judge $\mathcal{J}$ (Yang et al., 2025a) returns 1 when the generated explanation $\hat{e}_i$ is semantically equivalent to $e_i$. Finally, $\mathcal{S}_{\mathrm{ref}}$ is obtained by averaging $s_i$ over all samples.

**Phase II: Rectification Efficacy.** This phase evaluates the effectiveness of the rectification action $\hat{a}_i$ generated by a model. The action is executed by an external image editor to produce a rectified image $\mathbf{I}_i^{(t+1)}$. Then, the improvement is assessed using a VQA-based alignment function $\mathcal{V}(\mathbf{I}, Q) \in [0, 1]$, which applies an external MLLM to answer the annotated question set $Q_i$ for both the original and rectified images (detailed in Appendix F.2). We introduce the Rectification Score ($\mathcal{S}_{\mathrm{rect}}$) to quantify the gain. Crucially, instead of absolute improvement, we calculate the normalized improvement relative to the initial error, computed over misaligned samples:

$$\mathcal{S}_{\mathrm{rect}} = \frac{1}{N_{\mathrm{neg}}} \sum_{i:v_i=\text{False}} \frac{\mathcal{V}(\mathbf{I}_i^{(t+1)}, Q_i) - \mathcal{V}(\mathbf{I}_i^{(t)}, Q_i)}{1 - \mathcal{V}(\mathbf{I}_i^{(t)}, Q_i)}. \quad (3)$$

Here, $N_{\mathrm{neg}}$ denotes the number of misaligned samples, and a higher $\mathcal{S}_{\mathrm{rect}}$ indicates that a larger fraction of the initial discrepancy with respect to the target prompt is corrected.

## 3. Method

The preceding section introduced R³-Bench to evaluate model RVG capabilities. Evaluations on this benchmark (Tab. 1) reveal a critical misalignment in current MLLMs. Specifically, these models possess strong reasoning skills yet fail to translate them into effective image refinement. Motivated by this, we propose R³-Refiner, a reinforcement learning framework that leverages these reasoning capabilities as feedback to better accomplish RVG tasks. R³-Refiner can be seamlessly integrated with various MLLMs to enhance the generation quality of different T2I models.

### 3.1. R³-Refiner

R³-Refiner is a dual-stage framework that optimizes the complete R³ loop using Group Relative Policy Optimization (GRPO) (Shao et al., 2024), as illustrated in Fig. 4.

Specifically, for each training input $\langle P, \mathbf{I}^{(0)} \rangle$, the policy $\pi_\theta$ samples a group of $N$ trajectories $\{o_1, \ldots, o_N\}$. Consistent with the definitions in Sec. 2.1, each trajectory is parsed

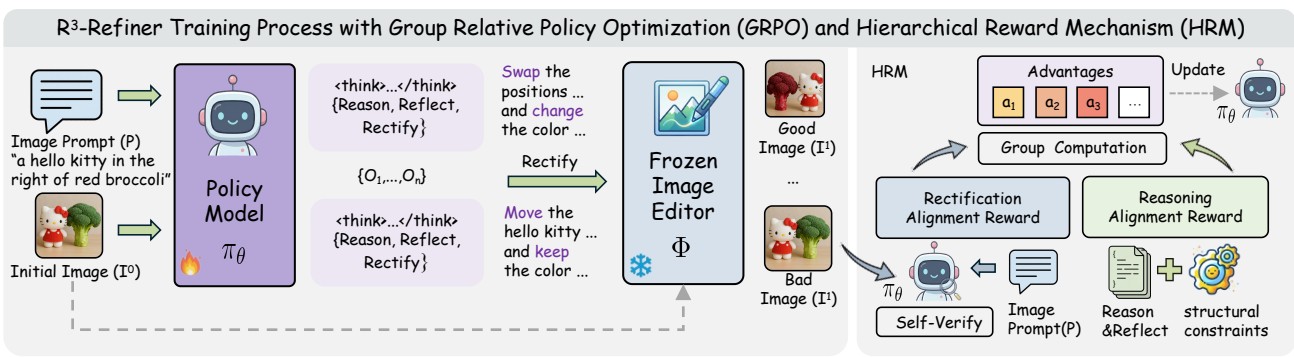

*Figure 4.* The policy $\pi_\theta$ samples $N$ structured trajectories via GRPO. The optimization is driven by a Hierarchical Reward Mechanism (HRM) comprising two stages: Reasoning Alignment ($R_{\text{reason}}$) and Rectification Alignment ($R_{\text{rect}}$).

as a tuple $o_i = \langle \hat{v}_i, \hat{e}_i, \hat{a}_i \rangle$, corresponding to the *Reason*, *Reflect*, and *Rectify* components, respectively. We implement the dual-stage optimization of these trajectories via the following Hierarchical Reward Mechanism (HRM).

**Hierarchical Reward Mechanism.** To facilitate image-text consistency verification, explanation generation, and visual rectification within the $R^3$ loop, we propose the Hierarchical Reward Mechanism (HRM), which integrates two complementary optimization signals: (i) the Reasoning Alignment Reward ($R_{\text{reason}}$), which aligns verification with GT labels, and (ii) the Rectification Alignment Reward ($R_{\text{rect}}$), which assesses consistency between the rectified image and the original prompt via $\pi_\theta$ itself. The design of HRM is motivated by our empirical observation that the model's discriminative capability significantly exceeds its visual rectification capability (Tab. 1). We detail the formulation of these rewards below.

**Stage I: Reasoning Alignment Reward.** This stage enhances the model's verification capabilities by targeting the *Reason* phase (verdict $\hat{v}$). While the *Reflect* phase (explanation $\hat{e}$) contains the reasoning chain, directly optimizing open-ended text generation via RL is unstable and computationally expensive. Following (Zhang et al., 2025b), we instead posit that the verdict $\hat{v}$ serves as a reliable proxy for the quality of the underlying reasoning. Consequently, we design $R_{\text{reason}}$ to enforce the accuracy of the final verdict against the GT $v_{\text{gt}}$ (derived from our data construction pipeline in Sec. 3.2):

$$R_{\text{reason}} = \lambda_{\text{fmt}} \cdot \mathbb{I}(o_i \in \Omega) + \lambda_{\text{acc}} \cdot \mathbb{I}(\hat{v}_i = v_{\text{gt}}). \quad (4)$$

Here, $\mathbb{I}(\cdot)$ denotes the indicator function. The coefficients $\lambda_{\text{fmt}}$ and $\lambda_{\text{acc}}$ weight the rewards for format compliance and prediction accuracy, respectively. The term $\Omega$ imposes format reward that encourages each trajectory $o_i$ to follow the prescribed template provided in Appendix F.1. The accuracy term, $\mathbb{I}(\hat{v}_i = v_{\text{gt}})$, weighted by $\lambda_{\text{acc}}$, constitutes the principal optimization objective, guiding the model to

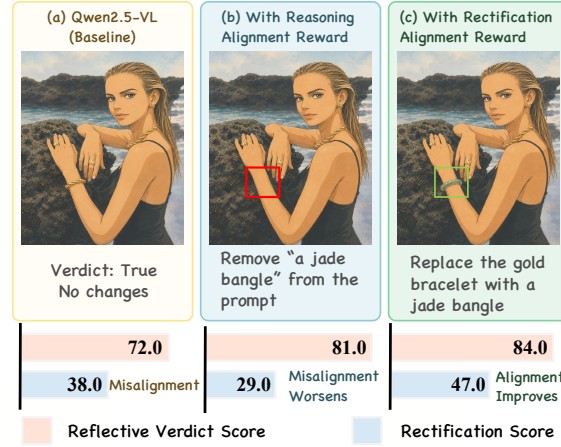

*Figure 5.* **Effectiveness of HRM.** The Reasoning Alignment Reward improves verdict accuracy but induces *Illusory Visual Rectification*. As shown in (**b**), the policy learns to edit the prompt rather than refining the image. The Rectification Alignment Reward in (**c**) alleviates this behavior and encourages valid visual rectification.

ground its judgments explicitly on accurate visual evidence. By enforcing correctness in the final *Reason* output ($\hat{v}$), we indirectly encourage logical consistency within the latent *Reflect* explanation ($\hat{e}$).

**Illusory Visual Rectification.** Stage I improves verification by aligning the *Reason* verdict with GT labels. A natural expectation is that stronger verification also leads to better rectification, because the policy should identify mismatches and then correct them. However, our empirical results, illustrated in Fig. 5(b), contradict this expectation. When trained exclusively with $R_{\text{reason}}$, the policy develops a shortcut behavior, improving rewards by rewriting the prompt description rather than genuinely rectifying the visual content. The policy edits the prompt instead of refining the image, which makes the pair appear consistent while the visual error persists. This behavior exposes a critical gap

between discriminative verification and constructive rectification. To address this issue, we introduce a second-stage reward that directly encourages effective visual rectification.

**Stage II: Rectification Alignment Reward.** In Stage II, the policy generates a rectification action $\hat{a}_i$ for a frozen editor $\Phi$ (Wu et al., 2025a). The editor executes $\hat{a}_i$ and produces the refined image $\mathbf{I}^{(1)}$. The policy then re-evaluates the consistency between the original prompt $P$ and $\mathbf{I}^{(1)}$. We define the Rectification Alignment Reward based on the policy's confidence in the consistency of the rectified pair:

$$R_{\text{rect}} = \mathbb{P}_{\pi_\theta}(\hat{v} = \text{True} \mid P, \mathbf{I}^{(1)}). \qquad (5)$$

By evaluating the rectified pair, $R_{\text{rect}}$ penalizes instruction-based shortcuts and encourages edits that resolve visual inconsistencies, as illustrated in Fig. 5(c).

Stage I enhances the verifier used to calculate $R_{\text{rect}}$ on $\mathbf{I}^{(1)}$ in Stage II, while Stage II leverages the enhanced verifier to provide execution-grounded supervision, thus promoting effective rectification actions.

**Iterative Refinement with $\text{R}^3$-Refiner.** After training, $\text{R}^3$-Refiner employs the $\text{R}^3$ loop for iterative refinement in T2I generation. As shown in Fig. 6, by repeatedly reasoning about image-text alignment and correcting localized errors, the framework progressively enhances image quality with each refinement step. The process continues until the policy confirms consistency or a predefined maximum number of iterations is reached. The effectiveness and generalization of $\text{R}^3$-Refiner have been validated across different models, as demonstrated in Tab. 1, 2, 3, and 5; iterative improvements are shown in Appendix C.4.

### 3.2. Scalable Paired Data Construction

To efficiently train $\text{R}^3$-Refiner, we develop a scalable data construction pipeline to obtain paired image-text data consisting of both aligned and misaligned samples, as illustrated in Fig. 9. Each sample is formatted as $\langle P, \mathbf{I}^{(0)}, v_{\text{gt}} \rangle$, where $P$ is the prompt, $\mathbf{I}^{(0)}$ denotes the image, and $v_{\text{gt}} \in \{\text{True}, \text{False}\}$ indicates if $P$ is consistent with $\mathbf{I}^{(0)}$.

**Multi-Source Synthesis Strategies.** To address multi-faceted alignment challenges, we curate a comprehensive dataset from diverse sources. To enhance generative quality, our *Generative Ranking* strategy employs a generate-and-rank paradigm based on prompts derived from T2I-R1 (Jiang et al., 2025). Candidate samples are assessed via T2I-CompBench (Huang et al., 2023), allowing us to select the top-$k$ ranked samples as positives and the bottom-$k$ as negatives, thus introducing distinct quality differences.

For achieving fine-grained alignment, we adopt the *Counterfactual Rewriting* strategy, which leverages high-quality pairs from BLIP-3O (Chen et al., 2025a). Original pairs

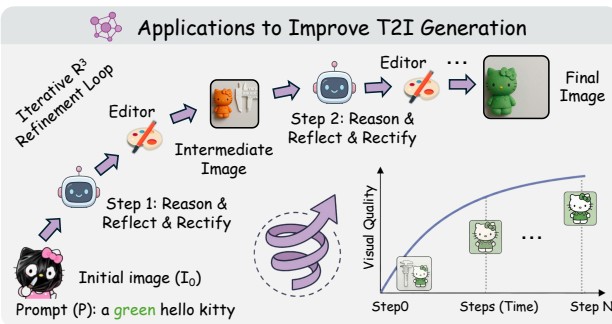

*Figure 6.* $\text{R}^3$-Refiner utilizes the iterative $\text{R}^3$ loop to continuously rectify image errors, allowing the final image quality to scale with the number of refinement steps.

are maintained as positive instances, while prompts undergo semantic alterations to intentionally contradict visual content, generating challenging negatives that necessitate precise grounding. Additionally, to simulate realistic application scenarios, we apply *Visual Inversion* to the PICO-Banana (Qian et al., 2025) dataset. Leveraging an MLLM (Bai et al., 2025b), we infer intended prompts based on editing differences, designating the successfully edited images as aligned examples and the pre-edit images as natural negatives.

**Cascaded Filtering.** To ensure label fidelity with minimal manual intervention, we introduce a three-stage verification pipeline for cascaded filtering. Specifically, the initial stage, *Rationale Verification*, serves as the primary filter employing a Proposer-Verifier mechanism where a specialized MLLM (Bai et al., 2025a) generates verdicts and explanations, which a generalist verifier (Zhang et al., 2025b) subsequently validates to exclude hallucinations lacking visual grounding or logical consistency.

Then, to further ensure reliability and mitigate model stochasticity, the second stage, *Consensus Voting*, involves repeated querying of a generalist model regarding object presence, quantity, and spatial arrangements. Only instances that consistently achieve high consensus across queries are retained. Finally, the third stage, *Visual Pruning*, leverages SAM3 (Carion et al., 2025) and CLIP-based scoring to filter out instances with ambiguous object boundaries or insufficient semantic alignment. Detailed dataset statistics and qualitative comparisons are presented in Appendix B.1. The data construction prompts are provided in Appendix F.3.

## 4. Experiments

In this section, we present the main results of $\text{R}^3$-Refiner on $\text{R}^3$-Bench and compare it against representative state-of-the-art verification and refinement methods. Subsequently, we demonstrate the plug-and-play capabilities of $\text{R}^3$-Refiner on general T2I benchmarks and analyze design choices through

*Table 1.* **Quantitative comparison on R³-Bench.** We report Reflective Verdict Score ($\mathcal{S}_{\text{ref}}$) and Rectification Score ($\mathcal{S}_{\text{rect}}$). We adopt Qwen-Image-Edit-2511 as the default editor for standard evaluations. Methods marked with † employ native editing modules. All results are evaluated by Qwen3-VL-235B. **Bold** indicates the best result within each group.

| Model / Method | Color | | Complex | | Non-Spa | | Numeracy | | Object | | Shape | | Spatial | | Texture | | Avg | |
|---|---|---|---|---|---|---|---|---|---|---|---|---|---|---|---|---|---|---|
| | $\mathcal{S}_{\text{ref}}$ | $\mathcal{S}_{\text{rect}}$ | $\mathcal{S}_{\text{ref}}$ | $\mathcal{S}_{\text{rect}}$ | $\mathcal{S}_{\text{ref}}$ | $\mathcal{S}_{\text{rect}}$ | $\mathcal{S}_{\text{ref}}$ | $\mathcal{S}_{\text{rect}}$ | $\mathcal{S}_{\text{ref}}$ | $\mathcal{S}_{\text{rect}}$ | $\mathcal{S}_{\text{ref}}$ | $\mathcal{S}_{\text{rect}}$ | $\mathcal{S}_{\text{ref}}$ | $\mathcal{S}_{\text{rect}}$ | $\mathcal{S}_{\text{ref}}$ | $\mathcal{S}_{\text{rect}}$ |
| GPT-4o (Hurst et al., 2024) | 0.77 | 0.64 | 0.72 | 0.30 | 0.80 | 0.48 | 0.78 | 0.41 | 0.87 | 0.87 | 0.74 | 0.54 | 0.72 | 0.41 | 0.71 | 0.53 | 0.76 | 0.53 |
| Banana | 0.84 | 0.67 | 0.78 | 0.31 | 0.78 | 0.07 | 0.90 | 0.49 | 0.92 | 0.78 | 0.83 | 0.39 | 0.88 | 0.48 | 0.80 | 0.55 | 0.84 | 0.50 |
| GPT-Image-1 | 0.82 | 0.68 | 0.80 | 0.39 | 0.78 | 0.33 | 0.74 | 0.52 | **0.96** | **1.00** | 0.74 | 0.46 | 0.84 | **0.63** | 0.76 | 0.58 | 0.79 | 0.57 |
| Gemini-3-Pro | 0.85 | **0.73** | 0.81 | 0.56 | 0.69 | 0.43 | **0.95** | 0.39 | **0.96** | 0.94 | 0.83 | 0.50 | **0.93** | 0.56 | 0.86 | 0.60 | **0.87** | 0.60 |
| GPT-5.2 | 0.82 | 0.71 | 0.63 | **0.57** | 0.61 | 0.86 | 0.91 | **0.61** | 0.92 | 0.96 | 0.71 | **0.56** | 0.89 | 0.55 | 0.84 | **0.63** | 0.80 | **0.65** |
| Bagel† (Deng et al., 2025) | 0.57 | 0.33 | 0.57 | 0.07 | 0.90 | 0.00 | 0.46 | 0.05 | 0.63 | 0.46 | 0.45 | 0.19 | 0.45 | 0.12 | 0.66 | 0.34 | 0.56 | 0.21 |
| OmniGen2† (Wu et al., 2025b) | 0.58 | -0.09 | 0.43 | -0.44 | 0.41 | 0.14 | 0.55 | -0.39 | 0.49 | 0.08 | 0.53 | -0.27 | 0.53 | -0.18 | 0.50 | -0.09 | 0.51 | -0.19 |
| ReasonEdit† (Yin et al., 2025) | 0.49 | 0.19 | 0.41 | -0.02 | 0.76 | 0.00 | 0.60 | 0.10 | 0.56 | 0.14 | 0.50 | 0.09 | 0.55 | -0.03 | 0.54 | 0.11 | 0.54 | 0.08 |
| SLD (Wu et al., 2024) | 0.44 | 0.17 | 0.21 | 0.16 | 0.14 | -0.24 | 0.51 | 0.34 | 0.28 | 0.16 | 0.24 | 0.18 | 0.57 | 0.52 | 0.39 | 0.41 | 0.37 | 0.28 |
| UniCot† (Qin et al., 2025) | 0.67 | 0.38 | 0.66 | 0.27 | 0.88 | 0.29 | 0.74 | 0.29 | 0.68 | 0.51 | 0.62 | 0.23 | 0.63 | 0.21 | 0.69 | 0.38 | 0.68 | 0.32 |
| ReflectionFlow† (Zhuo et al., 2025) | 0.67 | 0.18 | 0.77 | -0.03 | 0.80 | -0.07 | 0.77 | 0.28 | 0.85 | 0.61 | 0.80 | 0.06 | 0.79 | 0.32 | 0.80 | 0.45 | 0.78 | 0.26 |
| Reflect-DiT† (Li et al., 2025) | 0.73 | 0.25 | 0.53 | 0.25 | 0.92 | 0.57 | 0.51 | 0.37 | 0.82 | 0.76 | 0.64 | 0.08 | 0.75 | 0.39 | 0.64 | 0.53 | 0.68 | 0.37 |
| ThinkGen† (Jiao et al., 2025) | 0.78 | 0.47 | **0.84** | 0.07 | 0.90 | 0.14 | 0.92 | 0.32 | 0.93 | 0.65 | **0.86** | 0.25 | 0.89 | 0.38 | **0.89** | 0.43 | **0.87** | 0.37 |
| OmniVerifier (Zhang et al., 2025b) | 0.80 | 0.28 | 0.71 | -0.18 | 0.92 | 0.62 | 0.84 | 0.24 | 0.87 | 0.14 | 0.80 | 0.08 | 0.76 | 0.17 | 0.76 | 0.31 | 0.80 | 0.17 |
| Qwen2.5-VL-7B (Bai et al., 2025b) | 0.70 | 0.56 | 0.68 | 0.20 | 0.76 | 0.86 | 0.74 | 0.23 | 0.80 | 0.71 | 0.71 | 0.31 | 0.67 | 0.24 | 0.76 | 0.45 | 0.72 | 0.38 |
| + R³-Refiner (Ours) | **0.86** | 0.67 | 0.81 | 0.24 | **0.94** | 0.71 | 0.85 | 0.37 | 0.86 | 0.68 | 0.82 | 0.43 | 0.88 | 0.38 | 0.80 | 0.51 | 0.84 | 0.47 |
| Qwen3-VL-8B (Bai et al., 2025a) | 0.77 | 0.64 | 0.71 | 0.46 | 0.63 | 0.57 | 0.83 | 0.45 | 0.88 | 0.95 | **0.86** | 0.43 | 0.83 | 0.40 | 0.77 | 0.55 | 0.80 | 0.54 |
| + R³-Refiner (Ours) | 0.81 | 0.72 | 0.82 | 0.33 | 0.92 | **1.00** | 0.86 | 0.52 | 0.92 | 0.94 | **0.86** | **0.56** | 0.92 | 0.62 | 0.83 | 0.56 | **0.87** | 0.62 |

*Table 2.* **Quantitative results on GenEval++.** Each model uses the same model for initial generation and subsequent editing, except that Qwen-Image uses Qwen-Image for generation and Qwen-Image-Edit for editing. R³-Refiner performs verification and provides rectification instructions. All results are evaluated by Qwen3-VL-235B.

| Model | Color | Count | Col/Cnt | Col/Pos | Pos/Cnt | Pos/Size | Multi | Avg |
|---|---|---|---|---|---|---|---|---|
| SD-3-Med (Esser et al., 2024) | 0.550 | 0.500 | 0.125 | 0.350 | 0.175 | 0.150 | 0.225 | 0.296 |
| FLUX.1 (Labs, 2024) | 0.350 | 0.625 | 0.150 | 0.275 | 0.200 | 0.375 | 0.225 | 0.314 |
| Janus-Pro (Chen et al., 2025c) | 0.450 | 0.300 | 0.125 | 0.300 | 0.075 | 0.350 | 0.125 | 0.246 |
| Bagel (Deng et al., 2025) | 0.575 | 0.500 | 0.350 | 0.300 | 0.175 | 0.625 | 0.425 | 0.421 |
| + R³-Refiner | 0.650 | 0.650 | 0.500 | 0.400 | 0.300 | 0.675 | 0.550 | 0.532 |
| OmniGen2 (Wu et al., 2025b) | 0.625 | 0.250 | 0.150 | 0.300 | 0.100 | 0.475 | 0.300 | 0.314 |
| + R³-Refiner | 0.650 | 0.300 | 0.325 | 0.300 | 0.100 | 0.525 | 0.350 | 0.364 |
| Qwen-Image (Wu et al., 2025a) | 0.800 | 0.700 | 0.600 | 0.700 | 0.500 | 0.725 | 0.550 | 0.654 |
| + R³-Refiner | 0.925 | 0.775 | 0.775 | 0.725 | 0.550 | 0.700 | 0.550 | 0.714 |
| GPT Image | 0.925 | **0.900** | 0.825 | 0.625 | **0.600** | **0.875** | 0.800 | 0.793 |
| + R³-Refiner | **0.950** | 0.900 | **0.925** | 0.675 | 0.575 | **0.875** | 0.900 | **0.829** |
| Banana | 0.875 | 0.775 | 0.625 | 0.700 | 0.500 | 0.775 | 0.900 | 0.736 |
| + R³-Refiner | 0.925 | 0.850 | 0.725 | **0.775** | 0.575 | 0.775 | **0.975** | 0.800 |

*Table 3.* **Quantitative results on T2I-CompBench.** We follow the same generation and editing setup as in Tab. 2.

| Method | Color | Shape | Texture | Spatial | Complex | Avg |
|---|---|---|---|---|---|---|
| PixArt-α (Chen et al., 2023) | 0.669 | 0.493 | 0.648 | 0.206 | 0.343 | 0.472 |
| SD-v1.5 (Rombach et al., 2022) | 0.376 | 0.371 | 0.419 | 0.117 | 0.305 | 0.318 |
| SD-XL-base-1.0 (Podell et al., 2023) | 0.588 | 0.469 | 0.530 | 0.213 | 0.324 | 0.425 |
| FLUX.1 (Labs, 2024) | 0.741 | 0.572 | 0.692 | 0.286 | 0.370 | 0.532 |
| Janus-Pro (Chen et al., 2025c) | 0.636 | 0.353 | 0.494 | 0.206 | 0.356 | 0.409 |
| T2I-R1 (Jiang et al., 2025) | 0.813 | 0.585 | 0.724 | 0.338 | **0.399** | 0.572 |
| OmniGen2 (Wu et al., 2025b) | 0.776 | 0.516 | 0.709 | 0.393 | 0.371 | 0.553 |
| + R³-Refiner | 0.801 | 0.521 | 0.721 | **0.401** | 0.378 | 0.564 |
| Bagel (Deng et al., 2025) | 0.796 | 0.571 | 0.686 | 0.327 | 0.386 | 0.553 |
| + R³-Refiner | **0.842** | **0.603** | **0.739** | 0.357 | 0.398 | **0.588** |

Specifically, our method (built on Qwen3-VL-8B) attains an $\mathcal{S}_{\text{ref}}$ of 0.87, matching the performance of powerful closed-source models such as Gemini 3. In terms of rectification efficacy ($\mathcal{S}_{\text{rect}}$), while GPT-5.2 leads with 0.65, R³-Refiner yields a competitive 0.62, demonstrating that our RL-based optimization can effectively distill reasoning capabilities into effective rectification actions. Appendix C.3 further verifies the reliability of these comparisons with bootstrap and rank-stability analyses, and Appendix C.2 reports additional training-editor transfer results on R³-Bench under multiple inference-time editors.

ablation studies.

## 4.1. Benchmark Results

**Results on R³-Bench.** Tab. 1 reports the quantitative results on R³-Bench. We evaluate state-of-the-art (SOTA) models across the following categories: UMMs (Bagel, OmniGen2), MLLMs (Qwen2.5-VL, Qwen3-VL), existing RVG methods (SLD, ReasonEdit, UniCot, ReflectionFlow, Reflect-DiT, ThinkGen, OmniVerifier), and closed-source models (Gemini 3, GPT-4o, GPT-5.2, Banana, GPT-Image-1). R³-Refiner achieves SOTA performance among open-source methods.

**Results on T2I Benchmarks.** To assess the generalization capability of R³-Refiner, we evaluate its performance as a plug-and-play module on standard T2I benchmarks, including GenEval++ (Ye et al., 2025) and T2I-CompBench (Huang et al., 2023). We employ the iterative

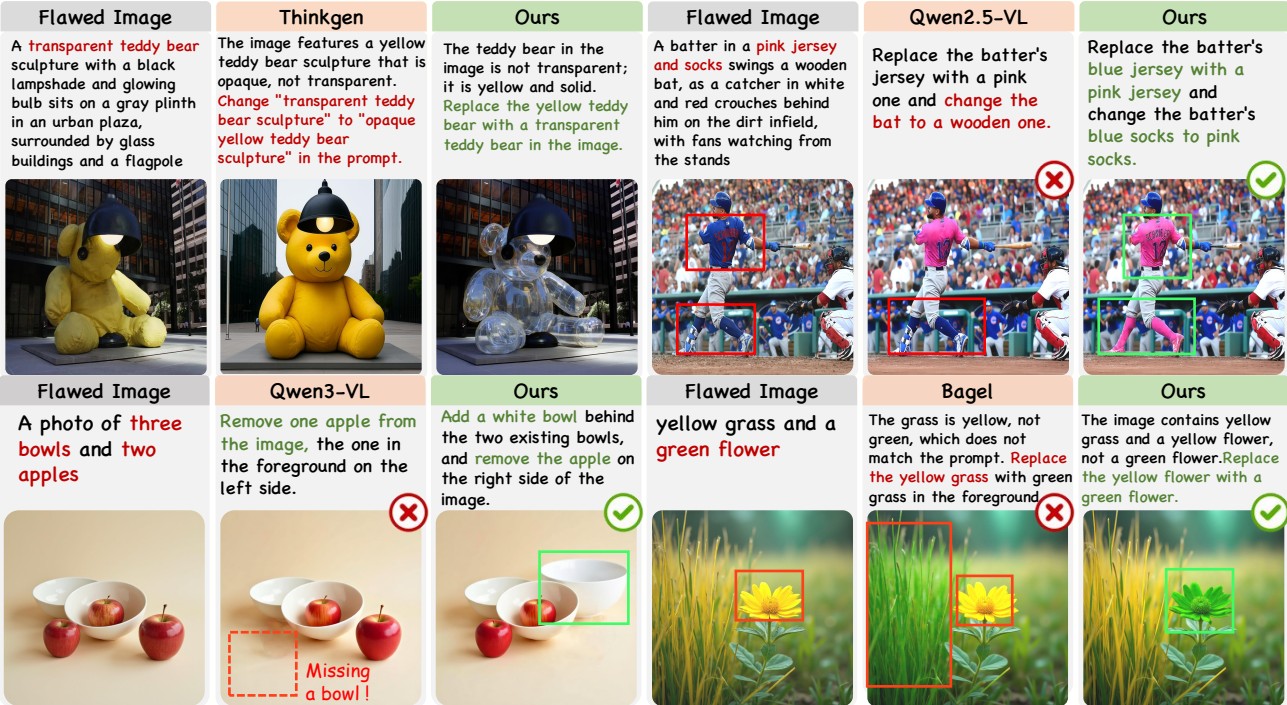

*Figure 7.* Qualitative comparison of R$^3$-Refiner with existing MLLMs, UMMs, and RVG methods.

*Table 4.* Ablation Study on Rectification Alignment Reward.

| Reward Mechanism | Reasoning ($S_{ref}$) | Rectification ($S_{rect}$) |
|---|---|---|
| Qwen2.5-VL-7B (Bai et al., 2025b) | 0.72 | 0.38 |
| Reasoning Reward Only ($R_{reason}$) | 0.81 (+0.09) | 0.29 (-0.09) |
| Hybrid ($R_{reason}$ + Decomposed QA) | 0.85 (+0.13) | 0.41 (+0.03) |
| Hybrid ($R_{reason}$ + SAM3 + CLIP) | 0.84 (+0.12) | 0.35 (–0.03) |
| HRM (Ours) | 0.84 (+0.12) | **0.47** (+0.09) |

*Table 5.* Comparison of R$^3$-Refiner and Best-of-$N$ on GenEval++.

| Model | N | Color | Count | Col/Cnt | Col/Pos | Pos/Cnt | Pos/Size | Multi | Avg |
|---|---|---|---|---|---|---|---|---|---|
| Qwen-Image (Wu et al., 2025a) | 0 | 0.800 | 0.700 | 0.600 | 0.700 | 0.500 | 0.725 | **0.550** | 0.654 |
| + Best-of-N (Bai et al., 2025b) | 3 | 0.875 | 0.700 | 0.725 | 0.700 | 0.550 | 0.725 | 0.500 | 0.682 |
| + Best-of-N | 4 | 0.825 | 0.725 | 0.625 | 0.725 | **0.575** | 0.725 | 0.500 | 0.671 |
| + Best-of-N | 5 | 0.850 | 0.725 | 0.625 | **0.750** | 0.550 | **0.750** | 0.550 | 0.686 |
| + Best-of-N | 6 | 0.825 | 0.725 | 0.600 | **0.750** | 0.575 | 0.725 | **0.550** | 0.679 |
| + R$^3$-Refiner | 1 | **0.925** | **0.800** | 0.650 | 0.725 | **0.575** | 0.700 | **0.550** | 0.703 |
| + R$^3$-Refiner | 2 | **0.925** | 0.775 | **0.775** | 0.725 | 0.550 | 0.700 | **0.550** | **0.714** |
| + R$^3$-Refiner | 3 | **0.925** | **0.800** | 0.650 | 0.700 | **0.575** | 0.700 | **0.550** | 0.700 |

refinement loop described in Sec. 3.1 with a maximum of two iterations. On GenEval++, R$^3$-Refiner consistently improves diverse base generators, including Bagel and Qwen-Image, and also enhances strong closed-source models such as Banana and GPT Image. For GenEval++, Appendix C.2 reports additional training-editor transfer results under multiple inference-time editors. We observe similar trends on T2I-CompBench, where R$^3$-Refiner improves the average scores of both OmniGen2 and Bagel. These results confirm that our policy improves visual generation through iterative refinement.

### 4.2. Ablation Study

**Rectification Alignment Reward Design.** The proposed R$^3$-Refiner is a two-stage reinforcement learning framework that utilizes GT labels for the first-stage reward. We investigate multiple alternatives for the second-stage reward design. Following (Jiang et al., 2025), we employ a CLIP-detector pipeline to generate fine-grained reward signals, substituting

the detector with SAM3. Additionally, we analyze question decomposition by partitioning prompt elements into subquestions and calculating individual rewards via VQA. As shown in Tab. 4, applying only the first-stage reward causes the model to exhibit illusory visual rectification (Sec. 3.1) and leads to rectification score degeneration. In contrast, the simplest image-text matching reward mechanism achieves optimal performance, demonstrating the effectiveness of the proposed self-reward paradigm.

**Comparison to Best-of-N.** To further enhance image generation quality, another strategy commonly employed during the inference phase is "Best-of-$N$". This strategy generates $N$ candidate images using different random seeds and subsequently utilizes an external evaluator to select the highest quality sample. This method improves quality at the cost of increased parallel computational overhead, contrasting with the serial optimization paradigm of R$^3$-Refiner. We compare the performance of R$^3$-Refiner against this strategy on the GenEval++ dataset, as presented in Tab. 5. Experi-

*Table 6.* **Human validation of $\mathcal{S}_{\text{rect}}$.** Human QA accuracy is compared with the automatic rectification score across representative models.

| Metric | GPT-5.2 | $R^3$-Refiner-BG | Gemini-3-Pro | Qwen3-VL-8B |
|---|---|---|---|---|
| $\mathcal{S}_{\text{rect}}$ | 0.650 | 0.657 | 0.599 | 0.543 |
| Human QA Acc. | 0.912 | 0.912 | 0.829 | 0.719 |

*Table 7.* **Iterative inference vs. learned policy on GenEval++.** All methods use Qwen-Image-Edit as the editor and report the average score.

| Method | Policy Source | $N=0$ | $N=1$ | $N=2$ |
|---|---|---|---|---|
| Qwen-Image | – | 0.654 | – | – |
| + Prompt resubmission | None | 0.654 | 0.668 | 0.654 |
| + Qwen3-VL-8B verifier | Pretrained | 0.654 | 0.639 | 0.571 |
| + $R^3$-Refiner-QE | RL, Qwen-Edit | 0.654 | 0.704 | **0.714** |
| + $R^3$-Refiner-BG | RL, Bagel | 0.654 | 0.686 | 0.711 |

mental results indicate that $R^3$-Refiner outperforms the peak performance of the Best-of-$N$ approach with a single RVG iteration. Furthermore, we observed a performance saturation phenomenon in both methods: as the value of $N$ increases, the performance of the Best-of-$N$ method does not improve significantly. Similarly, the performance of $R^3$-Refiner tends to saturate after two refinement iterations.

**Human Evaluation.** To validate the automatic rectification metric, we conduct a human study on 24 category-balanced $R^3$-Bench instances with 23 annotators. Annotators answer factual yes/no questions derived from the original prompts, and we compare the resulting human QA accuracy with $\mathcal{S}_{\text{rect}}$ over four representative models. $R^3$-Refiner-BG denotes the variant trained with Bagel (Deng et al., 2025). As shown in Tab. 6, human judgments preserve the same coarse ordering as the automatic metric, with GPT-5.2 and $R^3$-Refiner-BG tied at the top. The two rankings are strongly aligned (SROCC=0.800, KROCC=0.667), and annotators show consistent agreement (Fleiss' $\kappa = 0.776$). Additional evaluator-swap results are provided in Appendix C.1.

**Iterative Inference vs. Learned Policy.** To separate the effect of iterative editing from policy learning, we compare $R^3$-Refiner with two non-learned iterative alternatives under the same Qwen-Image-Edit editor (Wu et al., 2025a). Prompt resubmission re-feeds the original prompt and edited image to the editor without verification. $R^3$-Refiner-BG denotes the variant trained with Bagel (Deng et al., 2025), and the pretrained verifier uses Qwen3-VL-8B (Bai et al., 2025a). As shown in Tab. 7, prompt resubmission improves slightly at first but drops back by round two, suggesting that repeated editing without verification can corrupt already aligned content. The pretrained verifier degrades with iteration due to excessive false positives in verification. In contrast, only $R^3$-Refiner variants improve consistently across rounds, supporting that gains stem from the learned policy.

## 5. Conclusion

We propose $R^3$-Refiner to advance Reflective Visual Generation by addressing the misalignment where MLLMs accurately diagnose errors but fail to execute valid corrections. By incorporating a Hierarchical Reward Mechanism, our approach aligns the Iterative $R^3$ loop to facilitate precise and progressive visual refinement. Experiments on $R^3$-Bench, GenEval++, and T2I-CompBench demonstrate that our pol-

icy outperforms rigid verifiers and functions as a robust plug-and-play module for diverse generative executors. These findings highlight the value of Inference-Time Scaling for reliable visual synthesis.

## Acknowledgements

This work was supported by the Guangdong Basic and Applied Basic Research Foundation (2025A1515011546) and by the Shenzhen Science and Technology Program (JCYJ20240813105901003, ZDCY20250901113000001).

## Impact Statement

This paper introduces $R^3$-Bench and $R^3$-Refiner to improve the reliability of visual generative models by helping them diagnose and correct visual errors. Potential positive impacts include reducing misaligned generated content in creative, educational, and assistive applications. Potential risks include enabling more capable image-generation and editing systems that could be misused to create misleading synthetic content. We encourage deployment with provenance tracking, watermarking, access controls, and safeguards aligned with applicable policies.

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

# Supplementary Material

## A. Related Work

### A.1. Text-to-Image (T2I) Generation

T2I generation has advanced significantly. Leading diffusion models, such as Stable Diffusion (Esser et al., 2024; Podell et al., 2023) and FLUX (Labs, 2024; Labs et al., 2025), demonstrate impressive generative capabilities through large-scale training. Recent research shifts toward UMMs (Zeng et al., 2025; Tian et al., 2025; Wang et al., 2025a; Wu et al., 2025d;c; Geng et al., 2025; Huang et al., 2025b; Liu et al., 2025a; Liao et al., 2025; Wang et al., 2025d; Wu et al., 2025a; Zhao et al., 2025a; Xu et al., 2025; Chen et al., 2025a). These UMMs build upon the reasoning capabilities of MLLMs (Li et al., 2023; Team et al., 2025; Lu et al., 2024; Liu et al., 2023; Yang et al., 2025b; Li et al., 2024; Bai et al., 2025b;a; Wang et al., 2025b; Zhang et al., 2024; Zhu et al., 2023) and integrate multimodal understanding with generation into a unified architecture for controllable synthesis. Representative works include Emu (Cui et al., 2025), Show-o2 (Xie et al., 2025a), Janus-Pro (Chen et al., 2025c), OmniGen2 (Wu et al., 2025b), and Lumina-DiMOO (Xin et al., 2025). For instance, Bagel (Deng et al., 2025) and MMaDA (Yang et al., 2025d) utilize large-scale interleaved multimodal data and exhibit emergent capabili-

ties in complex generation and reasoning. Simultaneously, Z-Image (Cai et al., 2025) focuses on efficient native generation architectures. Despite these advancements, these models still struggle with compositional prompts as they operate in an open-loop and single-pass paradigm. This approach lacks mechanisms for error rectification and necessitates a transition to a multi-round RVG paradigm.

### A.2. Reasoning and Reflection in Visual Generation

Inspired by the success of self-reflection mechanisms in LLMs (Yao et al., 2022; Shinn et al., 2023; Ma et al., 2025; Huang et al., 2025d; Chen et al., 2025b; Sheng et al., 2025; Liu et al., 2025b; Su & Cardie, 2025; Shen et al., 2025; Yang et al., 2025c; Fang et al., 2025) and MLLMs (Ding & Zhang, 2025; Zhang et al., 2025a; Wang et al., 2025c; Kumar et al., 2024; Madaan et al., 2023; Lee et al., 2024), recent visual generation studies (Zou et al., 2025; Gu et al., 2025) explore reasoning generation and closed-loop paradigms. Several approaches (Jiao et al., 2025; Zeng et al., 2025; Yin et al., 2025) employ chain-of-thought reasoning to optimize input prompts and guide the image generation and editing process. ThinkMorph (Gu et al., 2025) investigates interleaved multimodal reasoning to align semantic understanding with visual synthesis. SLD (Wu et al., 2024) and OmniVerifier (Zhang et al., 2025b) serve as plug-and-play verifiers that detect and correct errors in image generation. Other strategies (Qin et al., 2025; Huang et al., 2025c; Zhuo et al., 2025; Li et al., 2025) utilize model-generated critiques to guide iterative refinements for enhanced semantic alignment and visual fidelity. However, we identify a critical capability misalignment where models fail to translate diagnostic reasoning into effective correction. Consequently, we propose a general reinforcement learning framework that aligns discriminative capabilities with actionable rectification by optimizing the entire Reason-Reflect-Rectify loop.

### A.3. Benchmarks for Visual Generation and Verification

Existing benchmarks primarily evaluate isolated capabilities within the domains of visual generation (Ye et al., 2025; Wei et al., 2025; Zhao et al., 2025b; Sun et al., 2025) and verification (Zhang et al., 2025b). For instance, T2I-CompBench (Huang et al., 2023; 2025a), GenEval (Ghosh et al., 2023), and DPG-Bench (Hu et al., 2024) target attribute alignment and compositional generation tasks, including spatial relationship modeling. Similarly, WISE (Niu et al., 2025) and KRIS (Wu et al., 2025e) assess the integration of world knowledge and commonsense reasoning into visual generation and editing. Furthermore, OneIG (Chang et al., 2025), MME-Unify (Xie et al., 2025b), and RealU-nify (Shi et al., 2025) introduce unified architectures cover-

ing understanding, generation, and multimodal tasks. However, these benchmarks predominantly focus on open-loop evaluation and fail to quantify the iterative reasoning integral to Reflective Visual Generation. To bridge this gap, we introduce $R^3$-Bench, which formalizes the Reason-Reflect-Rectify loop to assess the alignment between diagnostic reasoning and actionable rectification.

## B. Additional Qualitative Results

### B.1. Data Filtering Analysis

To evaluate the efficacy of the proposed Automated Cascaded Filtering pipeline (Fig. 9), we present a detailed statistical breakdown of the dataset composition alongside qualitative comparisons between rejected noise and the final high-quality data.

**Dataset Composition.** From an initial pool of approximately 40,000 synthesized samples, the three-stage filtering pipeline yielded a final $R^3$-Dataset of 24,925 high-fidelity instances. Fig. 8 depicts the hierarchical distribution of the curated dataset. This visualization details contributions from three distinct sources (inner ring), the diversity of fine-grained categories (middle ring), and the composition of preference pairs (outer ring). Specifically, the outer ring presents the distribution of aligned (positive) versus misaligned (negative) samples. This balanced structure facilitates model learning in distinguishing correct visual depictions from subtle hallucinations.

**Qualitative Quality Control.** Fig. 10 illustrates the efficacy of our quality control process. The top row displays rejected instances discarded due to critical deficiencies such as logical hallucinations (text contradicting image content), visual ambiguity, or segmentation artifacts. Conversely, the bottom row presents retained high-quality preference pairs that satisfy all verification criteria. These pairs feature distinct Aligned (Positive) and Misaligned (Negative) examples suitable for robust preference optimization.

### B.2. Extended Visualization of $R^3$-Bench

In this section, we present additional visualizations to illustrate the diversity of our benchmark and provide a qualitative comparison of $R^3$-Refiner against SOTA UMMs, MLLMs, and reflective visual generation methods.

**Visualizations of $R^3$-Bench.** As illustrated in Fig. 11, $R^3$-Bench is designed to cover a broad spectrum of visual challenges. The dataset spans eight fine-grained categories: *Color, Shape, Texture, Spatial, Numeracy, Object, Complex,* and *Non-Spatial*. Unlike existing benchmarks that focus on simple object existence, $R^3$-Bench includes "hard negatives" constructed via our counterfactual rewriting and

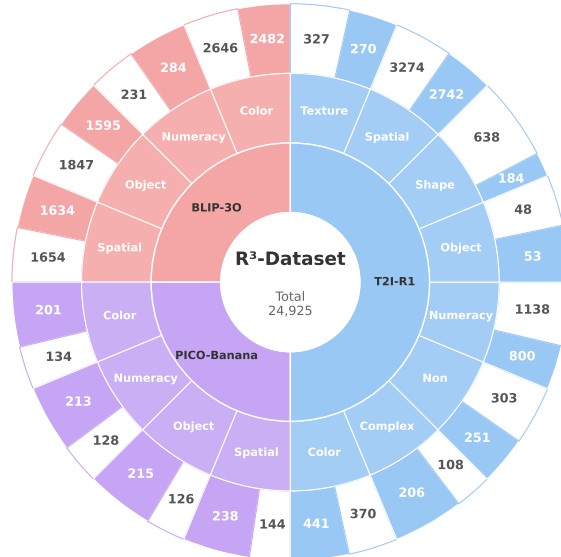

*Figure 8.* **Hierarchical Distribution of the $R^3$-Dataset.** The inner ring displays data sources, including T2I-R1, BLIP-3O, and PICO-Banana. Moving outward, the middle ring illustrates fine-grained error categories, such as Spatial, Color, and Numeracy. These categories are distributed evenly to maintain balanced visual diversity. Finally, the outer ring details the composition of the final preference pairs by indicating specific counts of Aligned (solid) and Misaligned (hollow) samples. The consistent presence of high-quality hard negatives alongside positives across all categories demonstrates the efficacy of the proposed data construction strategies.

visual inversion pipelines. For instance, the *Spatial* examples require precise understanding of relative positioning (e.g., "left of vs. right of"), while the *Numeracy* samples demand exact counting in cluttered scenes. This diversity ensures that $R^3$-Bench serves as a rigorous testbed for the complete Reason-Reflect-Rectify loop.

**Qualitative Comparison with SOTA Methods.** We provide a qualitative comparison between $R^3$-Refiner and varying baselines. As shown in the following figures, $R^3$-Refiner demonstrates superior capability across all three stages of the $R^3$ loop, effectively addressing common failure modes observed in existing methods.

*Type I: Verification Failures (Verdict Errors).* Fig. 12 illustrates the verdict stage. Baseline models often struggle with fine-grained visual discrimination. For instance, Bagel and ThinkGen frequently output incorrect "True" verdicts for mismatched images (e.g., missing objects or wrong colors), exhibiting a strong "yes-man" bias. Conversely, some methods like Reflect-DiT may hallucinate errors (False Negatives). $R^3$-Refiner accurately detects these subtle discrepancies, serving as a reliable gatekeeper.

*Type II: Hallucinated Reflections.* Fig. 13 highlights comparisons in the reasoning/explanation stage. Even when base-

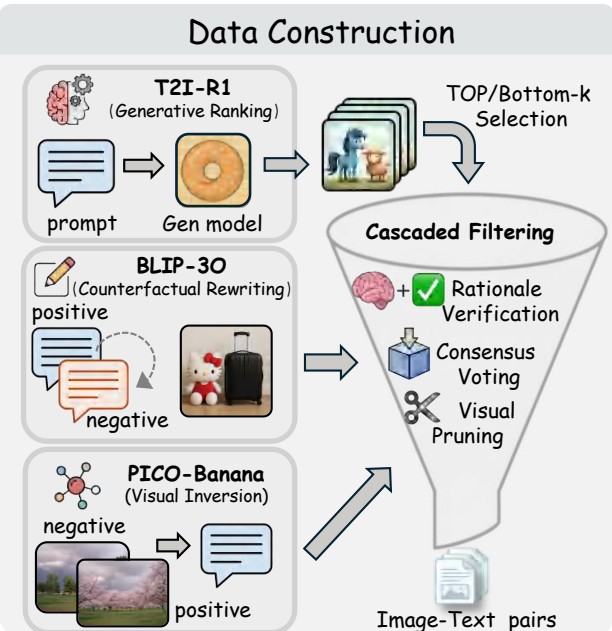

*Figure 9.* **Overview of the Scalable Paired Data Construction Pipeline.** The proposed pipeline is structured into two primary phases. The Multi-Source Synthesis Strategies phase initially leverages Generative Ranking, Counterfactual Rewriting, and Visual Inversion to synthesize diverse candidates with hard negatives. Subsequently, the Cascaded Filtering phase implements a three-stage verification mechanism comprising Rationale Verification, Consensus Voting, and Visual Pruning. This rigorous validation ensures high label fidelity and eliminates visual hallucinations. Ultimately, the pipeline yields high-quality aligned and misaligned sample pairs.

lines correctly identify an image as "False", their reasoning is often ungrounded. For example, ReasonEdit criticizes a specific object's color (e.g., "the hair dryer is black") even when the object is entirely missing from the image. $R^3$-Refiner avoids such hallucinations, providing explanations that strictly adhere to the visible pixel content.

*Type III: Evasive vs. Constructive Rectification.* Fig. 14 reveals a critical gap in the rectification stage. A pervasive issue with methods like OmniVerifier and ThinkGen is *Evasive Rectification*—they suggest modifying the user's text prompt to match the erroneous image (e.g., "Replace two bowls with two plates in the prompt") rather than fixing the image itself. $R^3$-Refiner, by contrast, generates constructive, actionable image editing instructions (e.g., "Replace the plates with bowls"), fulfilling the user's original intent.

### B.3. Failure Case Analysis

Despite its strong performance, $R^3$-Refiner faces challenges in extreme scenarios. As illustrated in Fig. 15, we identify two primary failure types: (1) Editor Capability Limits, where the policy generates a correct instruction (e.g., "add a

person"), but the backend editor fails to generate a realistic object; and (2) Dense Numeracy Errors, where the model occasionally miscounts objects in highly cluttered scenes (e.g., >10 items), likely due to the resolution constraints of the vision encoder.

## C. Additional Quantitative Analysis

### C.1. Evaluator Robustness

To assess the robustness of $S_{rect}$ to the choice of automated evaluator, we replace Qwen3-VL-235B (Bai et al., 2025a) with GPT-5.2 configured with low reasoning effort and re-run the $R^3$-Bench rectification evaluation without changing any other component of the pipeline. The comparison includes $R^3$-Refiner-BG trained with Bagel (Deng et al., 2025), GPT-4o (Hurst et al., 2024), Qwen-family MLLMs (Bai et al., 2025b;a), and existing verifier-based methods including SLD (Wu et al., 2024) and OmniVerifier (Zhang et al., 2025b). As shown in Tab. 8, the two evaluators produce highly consistent model rankings. The only minor discrepancy is between $R^3$-Refiner-BG and GPT-5.2, whose scores are nearly tied under both evaluators.

*Table 8.* **Evaluator robustness of $S_{rect}$.** We replace Qwen3-VL-235B (Bai et al., 2025a) with GPT-5.2 configured with low reasoning effort as the evaluator and report consistent rankings across representative models.

| Model | GPT Eval. | Qwen Eval. | GPT Rank | Qwen Rank |
|---|---|---|---|---|
| $R^3$-Refiner-BG | 0.62 | 0.66 | 1 | 1 |
| GPT-5.2 | 0.62 | 0.65 | 1 | 2 |
| $R^3$-Refiner-QE | 0.58 | 0.62 | 3 | 3 |
| Gemini-3-Pro | 0.55 | 0.60 | 4 | 4 |
| Qwen3-VL-8B | 0.50 | 0.54 | 5 | 5 |
| GPT-4o | 0.49 | 0.53 | 6 | 6 |
| Qwen2.5-VL-7B | 0.36 | 0.38 | 7 | 7 |
| SLD | 0.25 | 0.28 | 8 | 8 |
| OmniVerifier | 0.17 | 0.17 | 9 | 9 |

### C.2. Training-Editor Transfer

To examine whether $R^3$-Refiner transfers across training editors, we train two variants with different editors and evaluate them under multiple inference-time editors. $R^3$-Refiner-QE is trained with Qwen-Image-Edit, while $R^3$-Refiner-BG is trained with Bagel. As shown in Tab. 9, both variants consistently improve over their corresponding baselines across open-source and closed-source editors, supporting training-editor transfer.

### C.3. Benchmark Reliability

$R^3$-Bench is designed to evaluate whether models can diagnose semantically verifiable compositional errors and translate these diagnoses into effective rectification actions. It is not intended to exhaustively cover all generation fail-

| Category | Shape | Spatial | Numeracy | Color | Texture |
|---|---|---|---|---|---|
| Image Prompt | An oval rug and a square cushion | A banana above a bed | Two people and two camels | A white toothbrush and an orange horse | Wooden chopsticks and a glass window |

*Figure 10.* **Qualitative comparison of filtered noise and final data. Top:** Low-quality samples rejected by our pipeline due to hallucinations, ambiguity, or visual artifacts. **Bottom:** High-quality retained preference pairs. Each pair consists of an **Aligned** image (matching the prompt) and a **Misaligned** image (containing specific errors), providing the contrastive signal needed for training.

ures. Within this scope, we curate 670 expert-annotated test samples to enable controlled factual VQA evaluation while keeping human verification cost manageable. We assess whether this scale yields reliable and discriminative model comparisons through two complementary analyses.

**Paired Bootstrap.** First, we run paired bootstrap over the 670 test samples. We resample them with replacement for $B = 1000$ rounds using shared indices across models and seed 42, and compute 95% confidence intervals for pairwise $\mathcal{S}_{\text{rect}}$ differences between R$^3$-Refiner-BG and representative baselines. As shown in Tab. 10(a), R$^3$-Refiner-BG is statistically distinguishable from Gemini-3-Pro, Qwen3-VL-8B, and OmniVerifier at $\alpha = 0.05$, while its comparison with GPT-5.2 is non-significant. Given their small $\mathcal{S}_{\text{rect}}$ gap, this result is more consistent with a near-tie than with benchmark instability.

**Rank Stability.** We further evaluate rank stability by drawing stratified subsamples for 500 rounds at each sample size and computing Kendall's $\tau$ against the full-set ranking. As shown in Tab. 10(b), the ranking remains stable under subsampling, reaching $\tau = 0.95$ at $n = 400$. When the non-significant R$^3$-Refiner-BG/GPT-5.2 pair is treated as tied, the subsampled ranking exactly matches the full-set ranking at $n = 400$.

### C.4. Iterative Refinement Analysis

In this section, we explicitly analyze the iterative rectification capability of R$^3$-Refiner through a representative qualitative case study. As illustrated in Fig. 16, the refinement

*Table 9.* **Training-editor transfer.** We train R$^3$-Refiner with different editors and evaluate each variant under multiple inference-time editors on GenEval++ and R$^3$-Bench.

*(a) GenEval++ Avg*

| Method | GPT Image | Qwen-Image | OmniGen2 |
|---|---|---|---|
| Baseline | 0.793 | 0.654 | 0.314 |
| + R$^3$-Refiner-QE | 0.829 | 0.714 | 0.364 |
| + R$^3$-Refiner-BG | **0.839** | 0.711 | 0.343 |

*(b) R$^3$-Bench $\mathcal{S}_{rect}$*

| Method | Qwen-Edit | Banana | GPT Image |
|---|---|---|---|
| GPT-5.2 | 0.65 | 0.55 | **0.72** |
| Gemini-3-Pro | 0.60 | 0.56 | 0.71 |
| Qwen3-VL-8B | 0.54 | 0.49 | 0.63 |
| + R$^3$-Refiner-QE | 0.62 | **0.58** | **0.72** |
| + R$^3$-Refiner-BG | **0.66** | **0.58** | 0.71 |

*Table 10.* **Benchmark reliability analysis.** Paired bootstrap CIs and rank stability under subsampling. $\Delta\mathcal{S}_{rect}$ is computed as R$^3$-Refiner-BG minus the compared model.

*(a) Paired bootstrap CIs*

| Model | $\mathcal{S}_{rect}$ | CI Low | CI High | Sig. |
|---|---|---|---|---|
| GPT-5.2 | 0.653 | $-0.040$ | $+0.050$ | No |
| Gemini-3-Pro | 0.599 | $+0.008$ | $+0.109$ | Yes |
| Qwen3-VL-8B | 0.544 | $+0.064$ | $+0.163$ | Yes |
| OmniVerifier | 0.167 | $+0.427$ | $+0.555$ | Yes |

*(b) Rank stability*

| Metric | $n = 200$ | $n = 400$ | Full |
|---|---|---|---|
| Kendall's $\tau$ | 0.92 | 0.95 | 1.00 |
| Kendall's $\tau^{\dagger}$ | 0.97 | 1.00 | 1.00 |
| Exact Match | 87% | 100% | 100% |

*Table 11.* Detailed hyperparameter settings of R$^3$-Refiner preference optimization training.

| Hyperparameter | Value | Description |
|---|---|---|
| *General Optimization* | | |
| Optimizer | AdamW | With $\beta_1 = 0.9$, $\beta_2 = 0.999$. |
| Learning Rate | $1 \times 10^{-6}$ | With cosine decay scheduler. |
| Weight Decay | $1 \times 10^{-2}$ | L2 regularization coefficient. |
| Global Batch Size | 128 | Total batch size per update step. |
| Micro Batch Size | 4 | Per-device batch size for gradient accumulation. |
| Epochs | 5 | Total training epochs. |
| Max Prompt Length | 2560 | Maximum input tokens including image tokens. |
| Max Response Length | 2048 | Maximum generated output tokens. |
| *GRPO Algorithm* | | |
| Advantage Estimator | GRPO | Group Relative Policy Optimization. |
| Group Size ($N$) | 8 | Rollout samples per prompt for advantage estimation. |
| KL Coefficient ($\lambda_{kl}$) | $1 \times 10^{-2}$ | Weight for KL divergence penalty. |
| Clip Ratio | [0.2, 0.28] | Asymmetric PPO clipping range. |
| *Hierarchical Self-Rectification Rewards* | | |
| Stage-1 Weight ($\alpha_1$) | 0.25 | Weight for initial verification reward. |
| Stage-2 Weight ($\alpha_2$) | 0.75 | Weight for post-rectification reward. |
| Accuracy Weight ($\lambda_{acc}$) | 0.7 | Base reward for correct verification verdict. |
| Think Format Weight | 0.1 | Penalty for invalid thinking format ($\lambda_{fmt}$). |
| JSON Format Weight | 0.2 | Penalty for invalid JSON format ($\lambda_{fmt}$). |
| *Sampling Configuration* | | |
| Temperature (Train) | 1.0 | Exploration temperature during rollout. |
| Temperature (Eval) | 0.01 | Near-greedy decoding for evaluation. |
| Top-$p$ (Train) | 1.0 | No nucleus sampling truncation. |
| Top-$p$ (Eval) | 0.001 | Near-deterministic decoding. |
| *Data & Image Processing* | | |
| Rollout Batch Size | 128 | Batch size for generating rollouts. |
| Min Pixels | $512^2$ | Minimum image resolution. |
| Max Pixels | $\sim 1088^2$ | Maximum image resolution. |
| *Infrastructure* | | |
| Training Time | $\sim$3 days | Total wall-clock training duration. |

trajectory is visualized in three stages: the Left panel displays the initial image generated by the base model, which contains visual inconsistencies; the Middle panel presents the improved result after the first round of modification; and the Right panel shows the final output after the second round of modification, achieving full alignment with the target prompt.

## D. Training Implementation Details

### D.1. Implementation Hyperparameters

We utilize Qwen2.5-VL-7B-Instruct and Qwen3-VL-8B-Instruct as our base policy models $\pi_\theta$. The edit model is Qwen-Image-Edit-2511. The optimization is performed using the Group Relative Policy Optimization (GRPO) algorithm driven by the Hierarchical Reward Mechanism (HRM) defined in Sec. 3.1. The full training process takes approximately 3 days.

Tab. 11 lists the detailed hyperparameters. Note that the Stage weights ($\alpha_1$, $\alpha_2$) act as global scaling factors balancing the reasoning phase ($R_{reason}$) and rectification phase ($R_{rect}$). Within Stage I, the Accuracy Weight ($\lambda_{acc}$) and Format Weights ($\lambda_{fmt}$) specifically govern the trade-off between

verdict correctness and structural compliance.

## E. Evaluation Metrics Details

To comprehensively assess the performance of the R$^3$ pipeline, we introduce specific metrics aligned with the two-phase protocol defined in Sec. 2.3: Verdict-Reflection Alignment (Phase I) and Rectification Efficacy (Phase II). These metrics provide a rigorous evaluation by explicitly validating the correctness of the underlying reasoning process and quantifying the effective visual improvement relative to the error space.

### E.1. Phase I: Reflective Verdict Score ($\mathcal{S}_{ref}$)

The Reflective Verdict Score evaluates the fidelity of the model's diagnostic capability. Unlike simple binary classification metrics, $\mathcal{S}_{ref}$ imposes a strictly unified standard that penalizes "correct guesses" lacking valid reasoning.

**Metric Formulation.** The score $s_i$ for a single sample is calculated based on the ground truth verdict $v_i$.

*For Aligned Samples ($v_i = True$).* Since the image matches the prompt, no error explanation is required. The metric

degrades to a rule-based binary check:

$$s_i = \mathbb{I}(\hat{v}_i = \text{True})$$

*For Misaligned Samples ($v_i$ = False).* This is the critical evaluation scenario. Correctness requires the model to satisfy two conditions simultaneously: *verdict correctness*, where the model must correctly identify the mismatch ($\hat{v}_i$ = False), and *reasoning validity*, where the model's explanation $\hat{e}_i$ must be semantically equivalent to the ground truth diagnosis $e_i$. We verify the second condition using an LLM-Judge function $\mathcal{J}(e_i, \hat{e}_i)$ (see system prompt in Fig. 18, Appendix F.2). Thus:

$$s_i = \mathbb{I}(\hat{v}_i = \text{False}) \cdot \mathcal{J}(e_i, \hat{e}_i)$$

**Design Rationale.** This unified metric ensures that the model is not merely guessing the label but possesses a true comprehension of the visual discrepancies. By requiring explanation consistency for negative samples, we filter out spurious correctness.

### E.2. Phase II: Rectification Score ($\mathcal{S}_{\text{rect}}$)

The Rectification Score assesses the "action efficacy" of the model, specifically measuring the net gain in visual alignment after editing. We adopt a normalized formulation to rigorously quantify how much of the problem was solved.

**Metric Formulation.** We employ a VQA-based alignment function $\mathcal{V}(I, Q) \in [0, 1]$, which aggregates the verification results of atomic questions $Q_i$ decomposed from the prompt. The process involves three sequential steps:

*Decomposition.* The prompt $P_i$ is decomposed into atomic boolean questions $Q_i$ (see decomposition prompt in Fig. 19, Appendix F.2).

*Evaluation.* We calculate the alignment scores for both the original misaligned image ($S_{\text{pre}} = \mathcal{V}(I_i^{(t)}, Q_i)$) and the rectified image ($S_{\text{post}} = \mathcal{V}(I_i^{(t+1)}, Q_i)$) using the VQA verification prompt (see Fig. 20, Appendix F.2).

*Normalization.* Finally, the score represents the gain ($S_{\text{post}} - S_{\text{pre}}$) normalized by the maximum possible gain ($1 - S_{\text{pre}}$):

$$\mathcal{S}_{\text{rect}} = \frac{1}{N_{\text{neg}}} \sum_{i:v_i=\text{False}} \frac{\mathcal{V}(I_i^{(t+1)}, Q_i) - \mathcal{V}(I_i^{(t)}, Q_i)}{1 - \mathcal{V}(I_i^{(t)}, Q_i)} \quad (6)$$

**Design Rationale.** A "misaligned" input image is rarely 100% incorrect; it often partially matches the prompt (e.g., correct object but wrong color). Therefore, simply scoring the absolute quality of the final image is insufficient. We focus on measuring the *relative improvement*—the proportion of the previously unresolved error space that is successfully bridged by the model.

**Metric Interpretation.** The $\mathcal{S}_{\text{rect}}$ provides a distinct physical meaning regarding the editing quality: a score $> 0$ indicates valid visual improvement where the model successfully fixed errors; a score $\approx 1$ implies the error was completely resolved; conversely, a score $\leq 0$ denotes ineffective editing or degradation where the process introduced new errors.

## F. Prompt Details

In this section, we provide the exact prompt templates used in our R$^3$-Refiner framework and the baseline comparisons.

### F.1. Training Prompt for R$^3$-Refiner

Fig. 17 details the instruction employed by our policy $\pi_\theta$, designed to elicit the complete R$^3$ loop defined in Sec. 2.1. To facilitate complex reasoning, the prompt first requires the model to generate an internal chain-of-thought explicitly encapsulated within `<think>` tags. Subsequently, the model outputs the structured tuple $\langle v_t, e_t, a_t \rangle$ in a strict JSON format, where the components correspond to the `"answer"` (verification), `"explanation"` (reflection), and `"edit_prompt"` (rectification) fields, respectively.

### F.2. Evaluation Prompts

To ensure reproducibility, we provide the exact prompts used for the LLM-Judge ($\mathcal{J}$) in Phase I and the VQA-based alignment function ($\mathcal{V}$) in Phase II.

**Phase I: Verdict-Reflection Alignment.** Fig. 18 presents the system prompt used by the external LLM-Judge $\mathcal{J}$. This prompt is designed to evaluate the semantic equivalence between the generated reflection $\hat{e}_i$ and the ground truth explanation $e_i$, which is the core component of the Reflective Verdict Score ($\mathcal{S}_{\text{ref}}$).

**Phase II: Rectification Efficacy.** This phase quantifies the improvement of the rectified image using the Rectification Score ($\mathcal{S}_{\text{rect}}$). This process involves two steps: (1) decomposing the prompt into atomic questions, and (2) verifying these questions against the image.

*Question Decomposition:* To support fine-grained evaluation, we decompose the target prompt $P_i$ into a set of atomic Boolean questions $Q_i$. Fig. 19 presents the few-shot prompt used for this decomposition task.

*VQA-based Verification:* Fig. 20 displays the prompt template used for the VQA-based alignment function $\mathcal{V}$. This function applies an external MLLM to answer the decomposed questions $Q_i$, producing the probabilities used to calculate $\mathcal{S}_{\text{rect}}$.

### F.3. Data Construction Prompts

To ensure reproducibility of our data synthesis pipeline described in Sec. 3.2, we provide the exact prompts of data construction. We first present the generation prompts for Counterfactual Rewriting (Fig. 21) and Visual Inversion (Fig. 22), followed by the filtering prompts used in Rationale Verification (Fig. 23).

## G. Details of Test Set Curation

The construction of $R^3$-Bench follows a comprehensive four-stage pipeline designed to ensure high semantic diversity and annotation accuracy.

**Stage 1: Generative Data Sourcing.** We initially generate approximately 260,000 images using state-of-the-art text-to-image models (Deng et al., 2025; Wu et al., 2025a) based on prompts from T2I-R1 (Jiang et al., 2025) and GenEval++ (Ye et al., 2025). To efficiently identify valuable samples, we apply the Generative Ranking and Automated Cascaded Filtering pipeline proposed in this paper. This process filters the raw data into aligned and misaligned pairs based on image-text consistency. After splitting the data into training and testing sets, we obtain an initial set of 1,000 candidate samples from this generative stream.

**Stage 2: Real-world Data Augmentation.** To enhance domain diversity, we incorporate real-world image editing data from GEdit (Liu et al., 2025c). We first select English editing instructions and exclude categories unsuitable for visual reflection tasks (e.g., stylistic or background-only changes). Using the *gemini-2.5-flash-image* model, we generate the corresponding edited target images. To ensure image quality, we employ Qwen-VL to calculate the VIE-score (Liu et al., 2025c) and apply a Best-of-N selection strategy. We then synthesize misaligned samples by pairing the *pre-edit* source image with a caption of the *post-edit* image generated by Qwen-VL. This reverse-engineering approach contributes 300 additional challenging samples to the pool.

**Stage 3: Automated Annotation.** For the combined pool of 1,300 candidates, we employ advanced MLLMs to generate the necessary benchmark annotations. We use Qwen-VL to generate detailed explanations describing why the images deviate from the text prompts. Subsequently, we utilize *Qwen3-Next* to generate a set of Visual Question Answering (VQA) questions for each sample. These questions serve as the metric for evaluating the effectiveness of the rectification actions.

**Stage 4: Human Verification and Refinement.** To guarantee the gold-standard quality of $R^3$-Bench, human experts perform a final round of strict verification. Experts review the binary consistency labels to correct any automated judg-

ment errors. They also refine the generated explanations for clarity and verify the relevance of the VQA questions. Samples with ambiguous visuals or low-quality annotations are discarded. This rigorous human review results in the final set of 670 high-quality instances used in $R^3$-Bench.

## R³-Bench

### Color

**Prompt:** A photo of a white donut, a purple orange, and a pink tv remote.
**Answer:** False
**Explanation:** The TV remote in the picture is purple, not pink as mentioned in the prompt.
**Q1:** Is the TV remote pink in color?
…

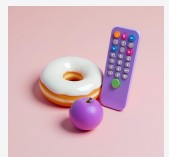

**Prompt:** a blue sun and a yellow sky.
**Answer:** False
**Explanation:** The image shows a yellow sun and blue sky, but the prompt requires a blue sun and yellow sky.
**Q1:** Is the sun blue in color?
…

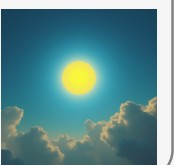

### Shape

**Prompt:** a spherical globe and a diamond map.
**Answer:** False
**Explanation:** The map is circular, not diamond-shaped as specified in the prompt.
**Q1:** Is the map diamond-shaped?
…

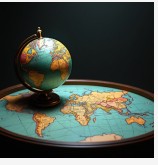

**Prompt:** a cylindrical vase and a rectangular flower arrangement.
**Answer:** False
**Explanation:** The vase is bulbous, not cylindrical, and the flower arrangement is round, not rectangular.
**Q1:** Is the vase cylindrical?
**Q2:** Is the flower arrangement rectangular?
…

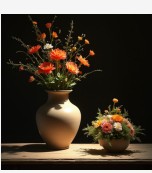

### Spatial

**Prompt:** a photo of a stop sign left of a bottle.
**Answer:** False
**Explanation:** The stop sign is right of the bottle in the image, but the prompt requires it…
**Q1:** Is the stop sign to the left of the bottle?
…

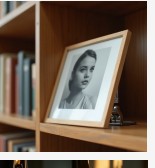

**Prompt:** a photo of a teddy bear right of a clock.
**Answer:** False
**Explanation:** The teddy bear is left of the clock, contradicting the prompt's requirement…
**Q1:** Is the stop sign to the left of the bottle?
…

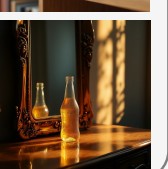

### Non-Spatial

**Prompt:** A child is playing with a toy car and racing it on a track.
**Answer:** False
**Explanation:** The surface is a smooth floor, not…
**Q1:** Is the toy car being raced on a track?
…

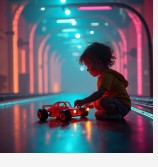

**Prompt:** A person is looking at a map of the universe and contemplating its vastness.
**Answer:** False
**Explanation:** The image shows a person looking at Earth (a planet), not a map of the universe as specified in the prompt.
**Q1:** Is the person looking at a map of the universe?
…

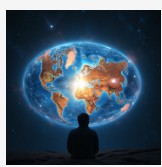

### Texture

**Prompt:** The plastic frame and metallic clip secure the glass photo on the bookshelf.
**Answer:** False
**Explanation:** The frame is wooden, not plastic as stated in the prompt.
**Q1:** Is the frame plastic?
…

**Prompt:** a plastic bottle and a glass mirror.
**Answer:** False
**Explanation:** The image shows a glass bottle instead of the specified plastic bottle.
**Q1:** Is the frame plastic?
…

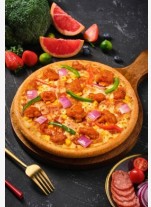

### Numeracy

**Prompt:** two kites and four birds.
**Answer:** False
**Explanation:** The image shows two birds instead of the four specified in the prompt.
**Q1:** Are there four birds in the image?
…

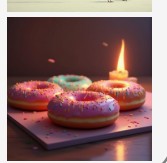

**Prompt:** a photo of three donuts.
**Answer:** False
**Explanation:** The image shows four donuts, contradicting the prompt's specification of three.
**Q1:** Are there exactly three donuts in the image?
…

### Object

**Prompt:** A round pizza with chicken, corn, peppers, onions, and a cherry sits on a wooden peel, surrounded by broccoli, grapefruit, strawberries, blueberries, limes, and a gold fork on a dark table.
**Answer:** False
**Explanation:** The image lacks a cherry on top of the pizza as specified in the prompt.
**Q1:** Is there a cherry on the pizza?
…

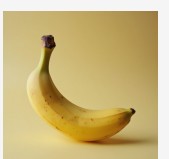

**Prompt:** a photo of a banana and a person.
**Answer:** False
**Explanation:** The image shows only a banana, omitting the person specified in the prompt.
**Q1:** Is there a person in the image?
**Q2:** Is there a banana in the image?
…

### Complex

**Prompt:** A photo of one tvs on the above and one zebras on the below.
**Answer:** False
**Explanation:** The zebra appeared inside the TV screen instead of being located at the bottom…
**Q1:** Is the TV positioned above the zebra?
…

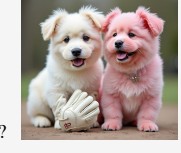

**Prompt:** A photo of two white baseball gloves and two pink dogs.
**Answer:** False
**Explanation:** The image shows only one white baseball glove, not the two mentioned in the prompt. The image also shows one white dog and one pink dog, not the two pink dogs mentioned in the prompt.
**Q1:** Is there exactly two baseball gloves in the image?
…

*Figure 11.* **Visualizations of R³-Bench.** The benchmark spans eight fine-grained categories (e.g., Spatial, Numeracy, Complex), designed to rigorously test visual reasoning and rectification.

| Input Image | Instruction | Results | Results *w/* Ours |
|---|---|---|---|

**Row 1:**
Instruction: a photo of a purple airplane and a pink toaster

Results (Bagel): **Answer:** True **Reflect:** Null **Rectify:** Null

Results w/ Ours: **Answer:** False **Reflect:** The pink object in the image is not a toaster; it resembles a vintage suitcase or toy vehicle, not a toaster. **Rectify:** Replace the pink suitcase-like object with a pink toaster that has a bread slot, lever, and typical toaster design.

**Row 2:**
Instruction: a photo of a blue dog and a yellow orange

Results (OmniGen2): **Answer:** True **Reflect:** The image accurately depicts a blue dog and a yellow orange, as specified in the prompt. The dog is positioned next to the orange, and both objects are correctly colored. **Rectify:** Null

Results w/ Ours: **Answer:** False **Reflect:** The orange in the image is orange-colored, not yellow as specified in the prompt. **Rectify:** Change the color of the orange to yellow by adjusting its hue to a yellow shade, ensuring it matches the 'yellow orange' description in the prompt.

**Row 3:**
Instruction: five giraffes roamed the safari

Results (OmniVerifier): **Answer:** True **Reflect:** The image accurately reflects the prompt by showing five giraffes standing in a line, which matches the description of 'five giraffes roamed the safari.' There are no discrepancies or errors in the image that contradict the prompt's statement. **Rectify:** Null

Results w/ Ours: **Answer:** False **Reflect:** The prompt specifies five giraffes, but the image contains only four giraffes. **Rectify:** Add a giraffe to the right of the rightmost giraffe, matching the size, pattern, and orientation of the existing giraffes.

**Row 4:**
Instruction: a photo of four pillows

Results (UniCoT): **Answer:** False **Reflect:** Null **Rectify:** Ensure there are four distinct, clearly visible pillows. Vary the pillow sizes, colors, and textures to show diversity. Make sure the textures are distinguishable and the background remains neutral to keep focus on the pillows.

Results w/ Ours: **Answer:** True **Reflect:** The image contains exactly four pillows, matching the prompt's requirement. **Rectify:** Null

**Row 5:**
Instruction: A woman is giving a presentation in a conference room.

Results (SLD): **Answer:** False **Reflect:** The image incorrectly contains the woman in the bottom-left which should be removed **Rectify:** Remove the woman in the bottom-left

Results w/ Ours: **Answer:** True **Reflect:** Null **Rectify:** Null

**Row 6:**
Instruction: a yellow grass and a green flower

Results (Reflect-DiT): **Answer:** False **Reflect:** There is no yellow grass in image. **Rectify:** There is no yellow grass in image

Results w/ Ours: **Answer:** True **Reflect:** Null **Rectify:** Null

**Row 7:**
Instruction: A photo of one brown bowls and two orange surfboards

Results (ThinkGen): **Answer:** True **Reflect:** The image contains one brown bowl (accurately matching "one brown bowl") and two surfboards (accurately matching "two surfboards"). Both surfboards are shades of orange (the left surfboard is a yellow-orange hue, and the right is a more vibrant orange, both falling within the "orange" color category specified in the prompt). **Rectify:** Null

Results w/ Ours: **Answer:** False **Reflect:** The prompt specifies two orange surfboards, but the image shows one yellow surfboard and one orange surfboard, so not both are orange. **Rectify:** Replace the yellow surfboard (left surfboard) with an orange surfboard to match the prompt's requirement of two orange surfboards.

*Figure 12.* **Qualitative comparison on Stage I (Verdict).** Baselines like Bagel and OmniGen2 often fail to detect semantic mismatches (e.g., identifying a pink toy car as a "pink toaster"). R³-Refiner correctly issues a "False" verdict based on precise visual evidence.

| Input Image | Instruction | Results | Results *w/* Ours |
|---|---|---|---|
| | a photo of a vase above an apple | **Answer:** False
**Reflect:** The vase is not above the apple; instead, the apple is inside the vase.
**Rectify:** Move the apple out of the vase and place it above it.

Bagel | **Answer:** False
**Reflect:** The image does not contain a vase above an apple; instead, it shows an apple with a flower growing from its stem.
**Rectify:** Add a glass vase above the apple, positioning it so the vase is clearly visible above the apple and the apple is below the vase.

Ours |
| | a brown banana and a green bird | **Answer:** False
**Reflect:** The image shows a bird with a banana as its body, which is not consistent with the prompt that mentions a brown banana and a green bird.
**Rectify:** Replace the bird with a green bird and the banana with a brown banana.

OmniGen2 | **Answer:** False
**Reflect:** The banana in the image is yellow, not brown as specified in the prompt.
**Rectify:** Replace the yellow banana with a brown banana, ensuring the banana's color is uniformly brown.

Ours |
| | A photo of two purple oranges and two green bananas | **Answer:** False
**Reflect:** The image shows two purple fruits that resemble oranges and a bunch of green bananas… The image accurately reflects this description with two purple fruits that resemble oranges and a bunch of green bananas. There are no errors in the description provided in the prompt.
**Rectify:** No changes are needed…

OmniVerifier | **Answer:** False
**Reflect:** The image contains more than two bananas, but the prompt specifies two green bananas.
**Rectify:** Remove the extra bananas from the bunch so that only two green bananas remain, keeping the two purple oranges in their original positions.

Ours |
| | a photo of a brown hair drier and a black stop sign | **Answer:** False
**Reflect:** The user requested a photo of a brown hair dryer and a black stop sign. The generated image incorrectly features a black hair dryer instead of a brown one,…
**Rectify:** Change the color of the black hair dryer to brown.

ReasonEdit | **Answer:** False
**Reflect:** The prompt mentions a brown hair drier, but the image contains only stop signs (a red 'STOP' sign and a black empty octagonal sign) with no hair drier present.
**Rectify:** Add a brown hair drier to the left side of the image, positioned next to the red stop sign.

Ours |
| | four helicopters and four hamburgers filled the hangar | **Answer:** False
**Reflect:** The image incorrectly contains the helicopter in the bottom-left, the helicopter in the center which should be removed. The image is missing a hamburger in the bottom, a hamburger in the bottom, …
**Rectify:** Remove the helicopter in the bottom-left, the helicopter in the center. Add a hamburger in the bottom…

SLD | **Answer:** False
**Reflect:** The image contains four helicopters but no hamburgers, contradicting the prompt's requirement of four hamburgers.
**Rectify:** Add four hamburgers on the concrete floor near the front left corner of the hangar, positioned next to the leftmost helicopter's landing gear.

Ours |
| | The fluffy white snow covered the rough brown ground. | **Answer:** False
**Reflect:** The ground is covered in snow, but it appears to be a forest floor with tree trunks and branches, not rough brown ground.
**Rectify:** Replace the tree trunks and forest background with rough brown ground, keeping the snow-covered foreground.

ReflectionFlow | **Answer:** False
**Reflect:** The prompt describes a 'rough brown ground' covered by snow, but the image shows snow without visible brown ground texture, indicating the ground's color and texture are not represented.
**Rectify:** Add a visible brown, rough ground texture beneath the snow in the lower-left corner of the image.

Ours |
| | A photo of one tvs on the above and one zebras on the below | **Answer:** False
**Reflect:** The prompt states "one tvs" (implying plural) and "one zebras" (implying plural), but the image contains one TV (singular) and one zebra (singular). The grammatical errors in the prompt…
**Rectify:** Correct the plural forms to singular: "A photo of one TV on the above and one zebra on the below"

ThinkGen | **Answer:** False
**Reflect:** The prompt states 'one zebras on the below', implying the zebra should be below the TV, but in the image, the zebra is displayed on the TV screen, not below it.
**Rectify:** Move the zebra image from the TV screen to the area below the TV stand, positioning it on the floor where the rug is.

Ours |

*Figure 13.* **Qualitative comparison on Stage II (Reflection).** Existing methods frequently hallucinate details in their explanations. For instance, **ReasonEdit** attempts to correct the color of a non-existent object. R[3]-Refiner correctly identifies the root cause (e.g., missing object) without fabrication.

| Input Image | Instruction | Results | Results *w/* Ours |
|---|---|---|---|

**Row 1 — Bagel / Ours**

Instruction: A photo of one carrot, two vases, and three beds

Results (Bagel):
**Answer:** False
**Reflect:** The image does not contain any beds.
**Rectify:** Add a bed in the background behind the vases and carrot.

Results *w/* Ours:
**Answer:** False
**Reflect:** The image contains three vases instead of two and no beds, while the prompt specifies two vases and three beds.
**Rectify:** Remove the middle vase (the tallest one) and add three beds in the background, positioned behind the remaining vases.

**Row 2 — OmniGen2 / Ours**

Instruction: a red apple and a green orange

Results (OmniGen2):
**Answer:** False
**Reflect:** The image shows a red apple and an orange, but the prompt specifically mentioned a green orange, …
**Rectify:** Replace the orange with a green orange in the image.

Results *w/* Ours:
**Answer:** False
**Reflect:** The orange in the image is orange, not green as specified in the prompt.
**Rectify:** Replace the orange on the right with a green orange.

**Row 3 — OmniVerifier / Ours**

Instruction: a photo of a blue couch and a red zebra

Results (OmniVerifier):
**Answer:** False
**Reflect:** The zebra in the image is black and white, not red, which contradicts the description in the prompt.
**Rectify:** Replace the description of the zebra as 'red' with 'black and white' in the prompt.

Results *w/* Ours:
**Answer:** False
**Reflect:** The zebra in the image is black and white, not red as specified in the prompt.
**Rectify:** Change the zebra's color to red by replacing its black and white stripes with red and white stripes.

**Row 4 — ReasonEdit / Ours**

Instruction: a photo of a brown fire hydrant

Results (ReasonEdit):
**Answer:** False
**Reflect:** The user instructed to change the color of the fire hydrant to brown. However, the resulting image is identical to the original image, with the fire hydrant still being red. The edit was not performed at all, indicating a complete failure to follow the instruction.
[Failed: Cannot be fixed by editing]
**Rectify:** Null

Results *w/* Ours:
**Answer:** False
**Reflect:** The fire hydrant in the image is red, not brown as specified in the prompt.
**Rectify:** Change the color of the fire hydrant from red to brown, adjusting all parts (body, top, caps) to brown.

**Row 5 — SLD / Ours**

Instruction: A boy in a blue shirt wears black headphones, typing on a silver laptop at a white table with a white cup and another laptop nearby

Results (SLD):
**Answer:** False
**Reflect:** The image incorrectly contains the laptop in the bottom-left which should be removed. The image is missing a boy in blue shirt with black headphones in the center as specified.
**Rectify:** Remove the laptop in the bottom-left. Add a boy in blue shirt with black headphones in the center

Results *w/* Ours:
**Answer:** False
**Reflect:** The image does not contain a boy, blue shirt, black headphones, or another laptop as specified in the prompt.
**Rectify:** Add a boy wearing a blue shirt and black headphones sitting at the white table in front of the laptop, and add another laptop to the left side of the table near the document stand.

**Row 6 — UniCoT / Ours**

Instruction: a blue bench and a green boat

Results (UniCoT):
**Answer:** False
**Reflect:** Null
**Rectify:** Add a green boat to the scene, positioned on or near a body of water to match the outdoor setting. Ensure the boat's color is a vibrant green to align with the prompt. Adjust the lighting and reflections to create a cohesive look with the existing elements.

Results *w/* Ours:
**Answer:** False
**Reflect:** The prompt mentions a green boat, but the image only contains a blue bench with no boat present.
**Rectify:** Add a green boat to the right side of the image, positioned next to the blue bench.

**Row 7 — ThinkGen / Ours**

Instruction: one frog, two bowls, one cat and two cakes

Results (ThinkGen):
**Answer:** True
**Reflect:** The prompt specifies "two bowls", but the image shows two cakes placed on plates (not bowls). While the image contains one frog, one cat, and two cakes, the container for the cakes (plates vs. bowls) does not match the prompt's requirement.
**Rectify:** Replace "two bowls" with "two plates" in the prompt to accurately describe the image.

Results *w/* Ours:
**Answer:** False
**Reflect:** The prompt specifies 'two bowls', but the image shows two plates under the cakes instead of bowls.
**Rectify:** Replace the two plates under the cakes with two bowls; position the bowls exactly where the plates are currently located.

*Figure 14.* **Qualitative comparison on Stage III (Rectification).** A common failure mode in baselines (e.g., ThinkGen, OmniVerifier) is proposing to edit the *text prompt* instead of the image. R³-Refiner generates specific image editing instructions (e.g., "Add a green boat") to align the visual content with the original prompt.

| Input Image | Instruction | Results |
|---|---|---|

**Answer:** False
**Reflect:** The image shows only two sofas and three guitars, whereas the prompt requires three sofas and four guitars.
**Rectify:** Add one more sofa on the right side of the image, mirroring the existing right sofa, and add one more guitar on the new sofa or on the existing right sofa to make a total of four guitars.

*three sofas and four guitars decorated the lounge*

*(a)* Visualization of failure case (Exmaple 1)

**Answer:** False
**Reflect:** The image shows only one phone (not two), only three apples (not four), and two strawberries (not one).
**Rectify:** Add one more phone to the right side of the image, place it next to the existing phone, and add one more apple to the background area behind the existing apples, and remove one strawberry from the foreground to leave only one strawberry.

*two phones, four apples, one cup and one strawberry*

*(b)* Visualization of failure case (Exmaple 2)

**Answer:** False
**Reflect:** The prompt requires three lemons, but the image shows five lemons. The flags are correctly present and positioned.
**Rectify:** Remove the two lemons on the far right side of the image, leaving only three lemons in the row.

*three flags and three lemons*

*(c)* Visualization of failure case (Exmaple 3)

**Answer:** False
**Reflect:** The image shows only five sheep, but the prompt requires seven sheep. The count is incorrect.
**Rectify:** Add two more sheep: one to the left of the leftmost sheep and one to the right of the rightmost sheep, to make a total of seven sheep.

*A photo of seven sheeps*

*(d)* Visualization of failure case (Exmaple 4)

*Figure 15.* **Visualization of failure cases.** We provide several visualizations of failure cases, corresponding to two primary failure types: 1) **Dense Numeracy Errors** (panels (a) and (b)), 2) **Editor Capability Limits** (panels (c) and (d)).

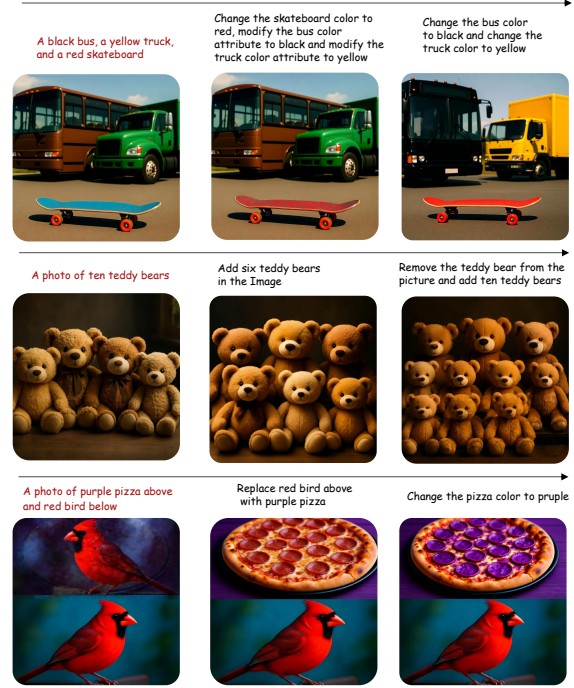

*Figure 16.* **Qualitative visualization of the iterative refinement process.**

---

**Training Prompt for R³-Refiner**

**User:** `<image>` This image was generated from the prompt: {{content | trim}}. Please carefully analyze the image and determine whether all the objects, attributes, count, and spatial relationships mentioned in the prompt are correctly represented in the image.

If the image accurately reflects the prompt, please answer "true"; otherwise, answer "false".

**When the answer is false, you must:**
1. Identify the main error(s) and describe them briefly in "explanation".
   - Clearly state what the prompt requires vs. what is actually shown in the image.
   - If there are multiple discrepancies (e.g., object missing, wrong color, wrong position, wrong count), you should mention all of the important ones in a concise way.
2. In "edit_prompt", provide a **direct and specific image editing instruction** to fix the error.
   - Choose the most appropriate action based on the actual error: add / remove / replace / move / change color / change shape / change texture / modify attribute / adjust count / resize / swap positions
   - The instruction can contain multiple coordinated edits in one sentence (e.g., change color + add object + move object), as long as it is clear and executable.
   - The instruction must specify the exact action and the location or reference point when relevant (e.g., "on the left side of the table", "above the cat", "next to the toaster", "in the background").
   - If the target position has no space, propose an alternative that is executable (e.g., swap positions, move the other object, resize first).

**Examples of good edit_prompt instructions:**
- "Replace the white candle on the right side of the image with a white candle holder."
- "Change the fork's color from silver to gold in the image."
- "Remove the single baseball glove from the left and add a kite in the sky on the left, and remove the kites from the right and add three baseball gloves to the ground on the right."
- "Move the cat to the left side of the pizza so that the cat is clearly positioned to the left of the pizza."
- "Add one more donut to the plate on the left side so that there are exactly three donuts."
- "Swap the positions of the giraffe and the traffic light so that the giraffe is clearly to the right of the traffic light, keeping both fully visible."

Respond strictly in the following JSON format:

```
{
    "answer": true/false,
    "explanation": "A brief, specific description of the main error (if answer is
  false).",
    "edit_prompt": "A concrete, location-specific editing instruction to fix the
  error (if answer is false)."
}
```

You should first think about the reasoning process in the mind and then provide the user with the answer. The reasoning process is enclosed within `<think> ... </think>` tags, i.e. `<think>` reasoning process here `</think>`answer here

---

*Figure 17.* The prompt used to train the R³-Refiner policy. This prompt enforces the iterative R³ loop (Sec. 2.1) by requiring the model to first generate an explicit internal reasoning process (CoT) before producing the structured tuple $\langle v_t, e_t, a_t \rangle$. These components are mapped to the JSON fields `"answer"` (corresponding to $v_t$, *Reason*), `"explanation"` (corresponding to $e_t$, *Reflect*), and `"edit_prompt"` (corresponding to $a_t$, *Rectify*), ensuring precise alignment with the formalized task definition.

---

**Phase I Evaluation: LLM-Judge for Verdict-Reflection Alignment ($\mathcal{J}$)**

**[System Prompt]**
You are an expert evaluator for image reflection tasks. Your task is to compare two explanations for why an image fails to match a prompt and determine if they are semantically equivalent.

You are given:
- The original image prompt.
- A Model Explanation and a GT (Ground Truth) Explanation.

Typical error dimensions include:
- **Color**: wrong or missing colors.
- **Object**: wrong, missing, or extra objects.
- **Numeracy**: wrong object counts or quantities.
- **Spatial**: wrong positions, relative locations, or spatial relations.
- **Shape**: wrong shapes or geometric properties.
- **Texture**: wrong material or surface appearance.
- **Complex**: complex combinations of multiple basic errors (for example, several objects and relations are all wrong at the same time, or multiple dimensions are intertwined).
- **Non**: more subjective or high-level mismatches that are not purely low-level visual attributes, such as incorrect actions or activities, scene type, atmosphere, style, or other semantic aspects that do not clearly fall into the categories above.

These are general categories that describe how an image can fail to match a prompt (for example: wrong color, wrong object, wrong count, wrong spatial relation, wrong shape or texture, wrong action or atmosphere, etc.).

**Definitions:**
- A "core error" is the main reason why the image does NOT satisfy the prompt (for example: wrong object count, wrong object type, wrong attribute, wrong spatial relation, missing or extra object, wrong action, wrong style, etc.).
- There can be multiple low-level details, but usually only a small number of core errors.

The model's explanation is considered correct if it identifies the SAME CORE ERROR as the GT (Ground Truth) explanation, even if:
- The wording is different
- Additional context or details are mentioned (for example, mentioning other objects that are present)
- The phrasing or style differs

**IMPORTANT:**
- Use the original prompt to understand what the image is supposed to contain or look like.
- Focus on whether both explanations point to the SAME fundamental problem in how the image fails to match the prompt, considering the typical dimensions listed above.
- Do NOT reject explanations just because one includes extra information or uses different words to describe the same error.
- However, if the Model Explanation introduces a NEW, SEPARATE core error that is not implied by the GT Explanation, or criticizes something that is actually correct according to the original prompt, then they are NOT semantically equivalent.

You should respond in JSON format:

```
{
    "is_correct": true/false,
    "reasoning": "A brief explanation of why the explanations are or are not
        semantically equivalent."
}
```

---

**[User Prompt]**
Original Prompt: {original_prompt}
Compare the following two explanations:
- Model Explanation: {model_explanation}

- GT Explanation: {gt_explanation}
Are these two explanations semantically equivalent? Respond in JSON format as specified.

---

*Figure 18.* The exact system prompt used by the LLM-Judge $\mathcal{J}$ in Phase I. This judge evaluates whether the generated reflection $\hat{e}_i$ is semantically equivalent to the ground truth explanation $e_i$, which is used to compute the Reflective Verdict Score ($\mathcal{S}_{\text{ref}}$).

---

**Question Decomposition Prompt (Phase II)**

Convert the given prompt into a JSON object containing a list of simple, verifiable boolean questions. The questions should focus on the prompt's main requirement, related to categories: Color, Shape, Texture, Spatial, Numeracy, Object, Complex, Non. For 'Non', generate questions that verify the main subjects, actions, and scene description.

**You MUST:**
- Focus on atomic facts (objects, attributes, relations, actions, counts).
- Make each question answerable as a boolean fact.
- Do NOT include any answers in the JSON.

---

**Example 1 (Numeracy):**
Input: "one rabbit and three horses" (numeracy)
```
{"yn_question_list": ["Is there a rabbit
in the image?", "Is there exactly one
rabbit?", "Are there horses in the image?",
"Are there exactly three horses?"]}
```

**Example 2 (Color):**
Input: "a gold bench and a green clock" (color)
```
{"yn_question_list": ["Is there a bench in
the image?", "Is the bench gold in color?",
"Is there a clock in the image?", "Is the
clock green in color?"]}
```

**Example 3 (Spatial):**
Input: "A horse on the right of a man" (spatial)
```
{"yn_question_list": ["Is there a horse
in the image?", "Is there a man in the
image?", "Is the horse to the right of the
man?"]}
```

**Example 4 (Shape):**
Input: "A circular chandelier..." (shape)
```
{"yn_question_list": ["Is there a
chandelier?", "Is the chandelier
circular?", "Is there a wall art?", "Is
the wall art triangular?"]}
```

**Example 5 (Texture):**
Input: "a plastic bottle and fabric pants" (texture)
```
{"yn_question_list": ["Is there a
bottle?", "Is the bottle plastic?", "Is
there a pair of pants?", "Are the pants
made of fabric?"]}
```

**Example 6 (Object):**
Input: "a surfboard and a knife" (object)
```
{"yn_question_list": ["Is there a
surfboard?", "Is there a knife in the
image?"]}
```

**Example 7 (Complex):**
Input: "The sweet chocolate chip..." (complex)
```
{"yn_question_list": ["Is there a
cookie?", "Is the cookie sweet?", "Is
there a crust?", "Is the crust crunchy?",
"Is there ice cream?", "Is the ice cream
soft?", "Did the cookie crumble on
them?"]}
```

**Example 8 (Non):**
Input: "A child is playing..." (non)
```
{"yn_question_list": ["Is there a
child?", "Is the child playing with a toy
airplane?", "Does the scene depict a child
making airplane noises?"]}
```

---

Now, perform the conversion for the following prompt:
Input Prompt: {original_prompt}    Input Category: {category}
Output JSON:

*Figure 19.* The few-shot prompt (8 examples) used to decompose user prompts into atomic boolean questions. To save space, the JSON outputs in the examples are displayed in a compact format; the actual prompt uses standard JSON indentation.

---

**Phase II Evaluation: VQA-based Alignment Function ($\mathcal{V}$)**

You are tasked with conducting a careful examination of the image. Based on the content of the image, please answer the following yes or no questions:
Questions: {questions}

**Note that:**
1. Each answer should be on a separate line, starting with "yes" or "no", followed by the reason.
2. The order of answers must correspond exactly to the order of the questions.
3. Each question must have only one answer.
4. Directly return the answers to each question, without any additional content.
5. Each answer must be on its own line!
6. Make sure the number of output answers equals the number of questions!

*Figure 20.* The prompt template used for the VQA-based alignment function $\mathcal{V}$ in Phase II. This prompt directs the external MLLM to answer the visual question set $Q_i$, producing the scores required to calculate the Rectification Score ($\mathcal{S}_{\text{rect}}$).

---

**Data Construction: Counterfactual Rewriting**

**System Instruction:** You are a highly precise image caption editor specialized in creating false captions for visual verification tasks.
**Your Task:**
- Given an original image caption (which correctly matches the image),
- you must modify it to create a **slightly but clearly incorrect** caption that would NOT match the original image anymore.

**First, analyze the caption type** (numeracy, color, texture, shape, spatial, object, complex, non). **Then, modify according to the caption type:**
1. **For numeracy**: Change the number (e.g., "four" → "two").
2. **For color**: Change one or more color attributes.
3. **For texture/shape**: Change material or shape attributes.
4. **For spatial**: Change spatial relationship (e.g., "above" → "below").
5. **For object**: Add/remove/replace an object.
6. **For complex/non**: Change attributes or action/verb context slightly.

**Strong requirements:**
1. Keep the **overall scene, entities, and structure** similar. Only change **1 or 2 local details**.
2. The change must be **specific and objectively checkable**.
3. The modification must be big enough to be false, but small enough to be plausible.

**Output format (JSON):**

```
{
  "false_prompt": "...",      // the modified caption that is now false
  "change_type": "...",       // e.g., "numeracy", "color"
  "changed_detail": "..."     // a short explanation of what changed
}
```

**Examples:** *Example (numeracy):* Original: "a photo of four coasters"

```
{"false_prompt": "a photo of two coasters", "change_type": "numeracy",
  "changed_detail": "Changed number from four to two."}
```

---

**User Input:** Original caption: {orig_caption}

*Figure 21.* The system prompt used to generate fine-grained hard negatives via counterfactual rewriting. The model rewrites a correct caption into a misaligned one by altering specific visual attributes.

---

**Data Construction: Visual Inversion**

**System Instruction:** You are an expert visual analyst capable of reverse-engineering image editing instructions.
**Input:** You are provided with two images: 1. **Pre-edit Image** (<image_1>): The original, misaligned image. 2. **Post-edit Image** (<image_2>): The successfully edited image that corrects a visual error.

**Your Task:** Compare the two images to identify the specific visual attribute that was modified. Based on this visual difference, infer the **Target Prompt** ($P$) that accurately describes the Post-edit Image.
**Critical Rules:**
- Focus ONLY on the visible content of the Post-edit Image.
- The generated prompt should imply the correction (e.g., if a red car became blue, the prompt should explicitly specify "a blue car").
- Identify the primary category of the change.

**Output Format (JSON):**

```
{
  "target_prompt": "Concise, factual description of the Post-edit Image",
  "change_category": "color|shape|texture|object|numeracy|spatial|non|complex",
  "confidence": "high"
}
```

*Figure 22.* The VLM prompt used for visual inversion. By comparing the pre-edit and post-edit images, the model infers the target prompt $P$ that aligns with the corrected visual state.

---

**Data Filtering: Rationale Verification**

**Step 1: Proposer Prompt (Initial Diagnosis)**

---

**System Instruction:** You should first think about the reasoning process in the mind and then provide the user with the answer. The reasoning process is enclosed within `<think> ...  </think>` tags, i.e., `<think> reasoning process here </think>` answer here.

**User Input:** <image> This image was generated from the prompt: {prompt}. Please carefully analyze the image and determine whether all the objects, attributes, and spatial relationships mentioned in the prompt are correctly represented in the image.
If the image accurately reflects the prompt, please answer 'true'; otherwise, answer 'false'. Respond strictly in the following JSON format:

```
{
    "answer": true/false,
    "explanation": "If the answer is false, briefly summarize the main error."
}
```

**Step 2: Verifier Prompt (Auditing)**

---

**System Instruction:** You are a principal evaluator reviewing a teacher model's (OmniVerifier) judgment on whether an image matches a text prompt. Your task is to verify if the teacher model's explanation is accurate and consistent with the actual image and prompt.

**Your review criteria:**
1. If the teacher model's answer is "true": Verify that the explanation correctly describes why the image matches the prompt, and that the described elements actually exist in the image.
2. If the teacher model's answer is "false": Verify that the explanation correctly identifies the actual problems, and that these problems truly exist in the image.

**Output Format:**

```
{
    "review_result": "pass" or "fail",
    "reasoning": "brief explanation..."
}
```

**User Input:** <image> **Prompt:** {prompt}
**Teacher Model's Answer:** {model_answer}
**Teacher Model's Explanation:** {explanation}

Please review:
- If answer is "true": Does the explanation correctly describe why the image matches? Do the described elements actually exist?
- If answer is "false": Does the explanation correctly identify the problems? Do these problems truly exist?

*Figure 23.* The prompts used in the Rationale Verification phase. The Proposer first generates a verdict with explicit reasoning (Step 1), and the Verifier audits the factual grounding of that explanation (Step 2) to filter hallucinations.

