# OpenReview forum: "Benchmarking and Evolving Reason-Reflect-Rectify for Reflective Visual Generation"
_ICML.cc/2026/Conference — ICML 2026 regular_

### Official Review · Reviewer_eRBv · 2026-03-06

**Soundness:** 2
**Presentation:** 3
**Significance:** 3
**Originality:** 3
**Overall Recommendation:** 4
**Confidence:** 5

**Summary:**

While UMMs and T2I models show impressive visual content generation capabilities, they still follow a single-turn generation while the multi-turn RVG is underexplored. In this work, the authors propose the R3Bench to diganose exsting models in reflective generation, the benchmark is mannually reviewed by human experts to ensure its quality. Furthermore, to improve the models in RVG, they introduce the R3Refiner and train it through online reinforcement learning with self-reward feedback. The experiments on Qwen2.5/3VL demonstrate the effectiveness of their R3Fefiner.

**Compliance With Llm Reviewing Policy:**

Affirmed.

**Final Justification:**

I appreciate the authors for their detailed response, which addressed my primary concerns regarding cross-editor evaluation. However, I suggest that the authors be cautious about framing $R^{3}$ as a major contribution (e.g., in Line 81 Right), as this is already a well-explored paradigm. The suggestions from other reviewers regarding the limited scale of the benchmark should also be taken into consideration. Overall, I have no further questions and will raise my score to 4.

**Key Questions For Authors:**

* Regarding Tab2 and 3, do all models utilize the same editor (Qwen-Image-Edit) to perform rectifications? If so, is the image generator used only to produce the initial image in the first round, with all subsequent images being produced by the editor?
* The authors should provide additional results using models more powerful than the editor, eg.nano-banana or GPT-Image. This is important to demonstrate that the improvements on GenEval++ and T2I-CompBench stem from the R3-Refiner, rather than the capabilities of the editor.
* Are stage 1 and stage 2 conducted independently? During stage 1, does the model generate 'rectify instructions'? Similarly, in stage 2, does it output the reason and reflect results? Furthermore, are these outputs included in the optimization process?

**Limitations:**

yes

**Strengths And Weaknesses:**

Strengths:
* The paper is well-written and easy to follow.
* The proposed R3-Bench is curated with human expert reviewing to ensure its quality, and provide finegrained evaluation of exsting models in RVG pipeline.
* The proposed R3-refiner is shown to be effective in imporving model performance on compositional text-to-image benchmarks, like Geneval++ and T2I-CompBench.

Weaknesses:
* The authors present R3-loop as one of their core contributions,  however, this paradigm has been extensively discussed in prior works [1][2].
* The evaluation of rectification relies on a single extra edit model. Consequently, the evaluation results become dependent on that specific editor's capabilities, leading to a bias toward the instruction space favored by that model. This makes it impossible to conduct a fair comparison of rectification quality.
* The authors should employ multiple editors for the evaluation on R3-Bench to determine whether the conclusions remain consistent across different editor models.
* Minor: L31, relective->reflective， L291, fig. 16->fig. 6


****
1. OmniGen2: Exploration to Advanced Multimodal Generation
2. Vision as a Dialect: Unifying Visual Understanding and Generation via Text-Aligned Representations

---

> ### Author Rebuttal · Authors · 2026-03-31
>
> We thank the reviewer for the constructive feedback. All your suggestions will be incorporated.
>
> > **Response to W1**
>
> **Distinction from Prior Work.** Our contribution is not the loop itself, but the analysis and mechanisms that make it effective (Table R1). [2] proposes “Self Reflect” as an offline SFT strategy (Appendix F) for image-prompt alignment judgment, without reflective visual generation (RVG). [1] attempts reflective generation via SFT but obtains $S_\\text{rect}=-0.19$ on R3-Bench (Table 1), indicating that reflection degrades output quality in that setting. Although [1] notes that RL may be needed, it leaves this as future work (Sec. 5.5). Neither work identifies the core issue in RVG. R3-Bench addresses this gap by decomposing RVG into three independently evaluable stages, showing that current models have strong diagnostic reasoning but weak rectification. Based on this finding, we identify Illusory Visual Rectification as a failure mode of naive RL reward design and propose R3-Refiner to address it, with cross-editor transfer across diverse editors and generators at inference time.
>
> **Table R1.** Comparison with cited reflective visual generation methods.
>
> | Feature | OmniGen2 [1] | Vision-Dialect [2] | R3-Refiner (Ours) |
> |---|---|---|---|
> | Reflective Visual Generation (RVG) | Yes | No | Yes |
> | Decoupled stage-wise evaluation of RVG | No | No | Yes (R3-Bench) |
> | Identified capability misalignment | Qualitatively | No | Yes (Table 1) |
> | Addressed Illusory Visual Rectification | No | No | Yes (Fig. 5) |
> | Cross-editor generalization | No | No | Yes (Tables 2, 3, R2, R3) |
>
> [1] *OmniGen2: Exploration to Advanced Multimodal Generation*, arXiv 2025.
> [2] *Vision as a Dialect: Unifying Visual Understanding and Generation via Text-Aligned Representations*, NeurIPS 2025.
>
> > **Response to W2, W3, Q1, and Q2**
>
> **Editor setup clarification.** For Table 1 (R3-Bench), all models use Qwen-Image-Edit as the unified editor for fair comparison. For Tables 2 and 3, each generator produces the initial image, after which R3-Refiner (trained with Qwen-Image-Edit) serves as the verifier and generates editing instructions, executed by each generator’s native editing capability (except Qwen-Image, which uses Qwen-Image-Edit). Thus, Tables 2 and 3 already form a cross-editor generalization setting. We will clarify this more explicitly in the revision.
>
> **Cross-editor generalization on GenEval++.** To further strengthen this point, we train a second R3-Refiner variant using BAGEL (Flux/MoT architecture) and evaluate both variants on GenEval++ across three generators, including GPT Image, which is stronger than the training editor. As shown in Table R2, all cross-editor settings improve over the corresponding baseline, supporting an editor-agnostic rectification policy.
>
>
>
> **Table R2.** Cross-editor generalization on GenEval++ (Avg, $N=2$). GPT Image and OmniGen2 use native editing at inference time; Qwen-Image uses Qwen-Edit. R3-Refiner-QE and R3-Refiner-BG are trained with Qwen-Edit and BAGEL, respectively.
>
> | Method | GPT Image | Qwen-Image | OmniGen2 |
> |---|---:|---:|---:|
> | Baseline | 0.793 | 0.654 | 0.314 |
> | + R3-Refiner-QE | 0.829 (+0.036) | 0.714 (+0.060) | 0.364 (+0.050) |
> | + R3-Refiner-BG | 0.839 (+0.046) | 0.711 (+0.057) | 0.343 (+0.029) |
>
> **Cross-editor evaluation on R3-Bench.** Following the reviewer’s suggestion, we also evaluate on R3-Bench using two additional editors, Banana and GPT Image. As shown in Table R3, both R3-Refiner variants consistently outperform the base model (Qwen3-VL-8B) across all editors, approaching or matching GPT-5.2 in several settings. This suggests that the gains come from R3-Refiner rather than from the capability of a particular editor.
>
> **Table R3.** Cross-editor evaluation on R3-Bench ($S_\\text{rect}$). Columns denote the inference-time editor. Deltas are relative to the base model (Qwen3-VL-8B).
>
> | Method | Qwen-Edit | Banana | GPT Image |
> |---|---:|---:|---:|
> | GPT-5.2 | 0.65 | 0.55 | 0.72 |
> | Gemini-3-Pro | 0.60 | 0.56 | 0.71 |
> | Qwen3-VL-8B (Base) | 0.54 | 0.49 | 0.63 |
> | R3-Refiner-QE | 0.62 (+0.08) | 0.58 (+0.09) | 0.72 (+0.09) |
> | R3-Refiner-BG | 0.66 (+0.12) | 0.58 (+0.09) | 0.71 (+0.08) |
>
> > **Response to Q3**
>
> Stage I and Stage II are jointly trained rather than optimized independently. For each input, the policy generates one sequence containing the verdict, explanation, and rectification instruction. Stage I rewards the verdict against the ground-truth label, while Stage II rewards the edited result after executing the predicted instruction with the frozen editor. These rewards are combined in a single GRPO objective, so the full sequence is optimized end-to-end. The explanation is not explicitly rewarded, but remains part of the optimized output.
>
> > **Response to Minor Comments**
>
> L31 “relective” will be corrected to “reflective,” and L291 “fig. 16” will be corrected to “fig. 6” in the revision.

---

> > ### Author Rebuttal · Reviewer_eRBv · 2026-04-02
> >
> > I appreciate the authors for their detailed response, which addressed my primary concerns regarding cross-editor evaluation. However, I suggest that the authors be cautious about framing $R^{3}$ as a major contribution (e.g., in Line 81 Right), as this is already a well-explored paradigm.
> >
> > Overall, I have no further questions and will raise my score to 4.

---

> > > ### Author Response · Authors · 2026-04-03
> > >
> > > Thank you for the positive acknowledgment and the constructive suggestion regarding the framing. We are glad that our response addressed your concerns, and we will refine the relevant wording in the revised version.

---

### Official Review · Reviewer_Zc4f · 2026-03-07

**Soundness:** 2
**Presentation:** 2
**Significance:** 2
**Originality:** 3
**Overall Recommendation:** 4
**Confidence:** 4

**Summary:**

This paper addresses the limitations of single-pass text-to-image models by formalizing a multi-round reflective visual generation framework called the Reason-Reflect-Rectify ($R^3$) loop. The authors introduce $R^3$-Bench, a benchmark with 670 expert-annotated instances to evaluate this specific iterative process. They also propose $R^3$-Refiner, a reinforcement learning method that uses Group Relative Policy Optimization and a Hierarchical Reward Mechanism to improve a model's ability to fix generated images. Experiments show that $R^3$-Refiner improves both reasoning and rectification scores and can act as a plug-and-play module to boost the performance of existing models.

**Compliance With Llm Reviewing Policy:**

Affirmed.

**Final Justification:**

The author's response addressed my concern and I will raise my score.

**Key Questions For Authors:**

Did you experiment with any other editing models besides Qwen-Image-Edit-2511 to verify the generalizability of the learned policy?

**Limitations:**

The authors do not discuss the limitations and potential negative societal impact of their work.

**Strengths And Weaknesses:**

Strengths：
1. The paper clearly identifies a critical gap in current models: they can accurately diagnose visual flaws but fail to execute valid corrections.
2. The $R^3$-Bench benchmark provides a specific way to evaluate the entire iterative Reason-Reflect-Rectify process, covering 8 distinct error categories
3.The Hierarchical Reward Mechanism successfully prevents the model from taking shortcuts, such as modifying the text prompt instead of actually refining the image content.
4. The proposed $R^3$-Refiner works effectively as a plug-and-play module that improves various base generators on standard benchmarks like GenEval++ and T2I-CompBench.

Weaknesses:
1. The benchmark size of exactly 670 instances is relatively small for comprehensive evaluation across diverse visual scenarios.
2. The framework relies heavily on a frozen external image editor to execute the rectification actions.
3. The overall visual improvement is strictly bounded by the inherent capabilities of the chosen external editor, which causes failures in extreme scenarios.

---

> ### Author Rebuttal · Authors · 2026-03-31
>
> Thank you for your valuable comments. All your suggestions will be incorporated.
>
> > **Response to W1**
>
> We address this concern with two analyses.
>
> **Bootstrap Confidence Intervals (CI).** Using paired bootstrap resampling [1], we resample the 670 instances with replacement ($B=1000$, seed=42), share indices across models, and compute 95% CIs for pairwise $\\Delta S_\\text{rect}$. As shown in Tab. R1, R3-Refiner is statistically distinguishable from Gemini-3-Pro, Qwen3-VL-8B, and OmniVerifier at $\\alpha=0.05$, while its comparison with GPT-5.2 is non-significant. Given their $S_\\text{rect}$ gap of only 0.005, this is more consistent with a near-tie than with insufficient benchmark sensitivity.
>
> **Rank Stability.** We further draw stratified subsamples (500 rounds per size) and compute Kendall's $\\tau$ against the full-set ranking. As shown in Tab. R2, $\\tau$ reaches 0.95 at $n=400$, with the only instability coming from the R3-Refiner/GPT-5.2 pair. This is expected given their 0.005 gap and the non-significant result in Tab. R1. Our benchmark size (670) is comparable to GenEval (553) and Step1X-Edit (606).
>
> **Tab. R1.** Paired bootstrap significance test for R3-Refiner-BG ($S_\\text{rect}=0.658$) vs. other models ($B=1000$, 95% CI of $\\Delta S_\\text{rect}$). Sig.: CI excludes zero.
>
> | Other Models | $S_\\text{rect}$ | 95% CI ($\\Delta$) | Sig. |
> |---|---:|---:|---:|
> | GPT-5.2 | 0.653 | [−0.040, +0.050] | ✗ |
> | Gemini-3-Pro | 0.599 | [+0.008, +0.109] | ✓ |
> | Qwen3-VL-8B | 0.544 | [+0.064, +0.163] | ✓ |
> | OmniVerifier | 0.167 | [+0.427, +0.555] | ✓ |
>
> **Tab. R2.** Rank stability under subsampling. $\\dagger$: treating the non-significant R3-Refiner/GPT-5.2 pair as tied.
>
> | Metric | n=200 | n=400 | Full |
> |---|---:|---:|---:|
> | Kendall $\\tau$ | 0.92 | 0.95 | 1.00 |
> | Kendall $\\tau^\\dagger$ | 0.97 | 1.00 | 1.00 |
> | Exact Match$^\\dagger$ | 87% | 100% | 100% |
>
> [1] *Statistical Significance Tests for Machine Translation Evaluation*, EMNLP 2004.
>
> > **Response to W2, W3 & Q1**
>
> **Cross-Editor Generalization.** Tabs. 2 and 3 in our submission already show that R3-Refiner trained with Qwen-Image-Edit generalizes to multiple editors at inference time. To further verify this, we train a second variant using BAGEL (Flux/MoT architecture) and evaluate both across multiple inference editors on GenEval++ (Tab. R3) and on R3-Bench with two additional editors, Banana and GPT Image (Tab. R4). All configurations show consistent improvement, supporting cross-editor generalization.
>
> **Editor Capability Upper Bound.** While editor capability constrains the achievable quality, R3-Refiner provides stable improvement across both open-source editors (OmniGen2, Qwen-Image-Edit) and strong closed-source editors (GPT Image, Banana), as shown in Tabs. R3 and R4. To further verify that the gains come from the learned policy rather than iterative inference, we perform an ablation under a fixed editor (Tab. R5): Best-of-N parallel sampling, prompt resubmission (re-feeding the original prompt and image to the editor), and a pretrained Qwen3-VL-8B verifier. Only the R3-Refiner variants improve consistently, suggesting that a learned policy is necessary.
>
> **Tab. R3.** Cross-editor generalization on GenEval++ (Avg, $N=2$). GPT Image and OmniGen2 use native editing at inference. Qwen-Image uses Qwen-Edit. R3-Refiner-QE and R3-Refiner-BG denote variants trained with Qwen-Edit and BAGEL, respectively.
>
> | Method | GPT Image | Qwen-Image | OmniGen2 |
> |---|---:|---:|---:|
> | Baseline | 0.793 | 0.654 | 0.314 |
> | + R3-Refiner-QE | 0.829 | 0.714 | 0.364 |
> | + R3-Refiner-BG | 0.839 | 0.711 | 0.343 |
>
> **Tab. R4.** Cross-editor evaluation on R3-Bench ($S_\\text{rect}$). Deltas are relative to the base model (Qwen3-VL-8B).
>
> | Method | Qwen-Edit | Banana | GPT Image |
> |---|---:|---:|---:|
> | GPT-5.2 | 0.65 | 0.55 | 0.72 |
> | Gemini-3-Pro | 0.60 | 0.56 | 0.71 |
> | Qwen3-VL-8B | 0.54 | 0.49 | 0.63 |
> | R3-Refiner-QE | 0.62 (+0.08) | 0.58 (+0.09) | 0.72 (+0.09) |
> | R3-Refiner-BG | 0.66 (+0.12) | 0.58 (+0.09) | 0.71 (+0.08) |
>
> **Tab. R5.** Iterative ablation on GenEval++. All methods use Qwen-Image-Edit as the editor.
>
> | Method | N | Color | Count | ... | Multi | Avg |
> |---|---:|---:|---:|---:|---:|---:|
> | Qwen-Image | 0 | 0.800 | 0.700 | ... | 0.550 | 0.654 |
> | + Best-of-N | 5 | 0.850 | 0.725 | ... | 0.550 | 0.686 |
> | + Prompt Resub. | 2 | 0.775 | 0.725 | ... | 0.575 | 0.654 |
> | + Qwen3-VL | 2 | 0.675 | 0.650 | ... | 0.525 | 0.571 |
> | + R3-Refiner-QE | 2 | 0.925 | 0.775 | ... | 0.550 | 0.714 |
> | + R3-Refiner-BG | 2 | 0.850 | 0.750 | ... | 0.675 | 0.711 |
>
> > **Response to Limitations**
>
> We will add this to the Limitations section. Rectification quality is bounded by the capability of the frozen editor, as discussed in Appendix B.3. In addition, R3-Bench currently focuses on compositional errors and does not yet cover stylistic or aesthetic dimensions. We will also discuss potential societal risks, including misuse for generating deceptive visual content.

---

> > ### Author Rebuttal · Reviewer_Zc4f · 2026-04-04
> >
> > N/A

---

> > > ### Author Response · Authors · 2026-04-04
> > >
> > > Thank you for the positive acknowledgment and for increasing your score. We are glad that our response addressed your concerns. We will incorporate the clarifications and new results into the revised paper.

---

### Official Review · Reviewer_bbzb · 2026-03-07

**Soundness:** 2
**Presentation:** 3
**Significance:** 2
**Originality:** 2
**Overall Recommendation:** 3
**Confidence:** 4

**Summary:**

This paper addresses iterative text-to-image generation through a critique-and-revise framework termed the Reason-Reflect-Rectify (R3) loop. The authors introduce R3-Bench, an evaluation set of 670 annotated image-text pairs, to measure a multimodal model's ability to diagnose visual errors and generate corresponding editing instructions. To train models for this task, they propose R3-Refiner, an RL pipeline utilizing Group Relative Policy Optimization (GRPO). The optimization relies on a dual-stage Hierarchical Reward Mechanism that penalizes format violations and measures visual alignment after an external, frozen image editor executes the policy's instructions.

**Compliance With Llm Reviewing Policy:**

Affirmed.

**Key Questions For Authors:**

1. How do you isolate a policy failure (generating a poor editing instruction) from an editor failure (failing to execute a valid instruction) during GRPO training? Equation 5 currently penalizes the policy for the frozen editor's shortcomings.
2. Given the high variance of diffusion models, how do you justify the statistical significance of evaluating 8 distinct compositional categories with only 670 total samples in R3-Bench?
3. How does the trained policy perform when the underlying editor (Φ) is swapped (e.g., to InstructPix2Pix or Flux.1) during inference? Evaluating this is necessary to prove the policy learned general rectification logic rather than editor-specific prompt engineering.
4. What is the measured false-positive and false-negative rate of the automated cascaded filtering pipeline used to curate the 24,925 training instances?

**Limitations:**

The authors briefly note "Editor Capability Limits" in Appendix B.3, but they do not address how this limitation fundamentally confounds their RL reward formulation. They entirely omit discussions regarding the statistical limitations of benchmarking 8 categories with only 670 samples.

**Strengths And Weaknesses:**

The paper strives to study a relevant question regarding the empirical gap between a model's ability to detect visual errors and its ability to correct them. The identification of "Illusory Visual Rectification" is an accurate observation of RL reward hacking. The authors correctly note that policies often learn to edit the text prompt to match the flawed image rather than fixing the image itself.

The methodology contains a fundamental RL credit assignment flaw. The pipeline relies entirely on a frozen external image editor (Qwen-Image-Edit) to execute the policy's textual instructions. The Stage II reward in Equation 5 is computed based on the visual alignment of the editor's output. If the policy generates a correct and highly specific editing instruction but the frozen editor fails to render it accurately, the policy receives a negative reward. The authors acknowledge editor capability limits in Appendix B.3 but fail to recognize that this confounds the mathematical foundation of their RL training. The metric permanently entwines the policy's reasoning capability with the editor's generative limits.

Overall, the authors present a critical issue regarding iterative generation, but the proposed evaluation benchmark is severely underpowered. R3-Bench contains 670 instances divided across 8 fine-grained categories, yielding approximately 83 samples per category. This sample size is statistically insufficient to evaluate open-ended compositional generation. Modern benchmarks require thousands of instances to account for the high variance in diffusion outputs and the extreme diversity of spatial and numerical compositions.

The originality of the submission is low. The literature on multimodal self-correction and LLM-as-a-judge pipelines is saturated. The proposed R3 loop is functionally identical to established critique-and-revise frameworks like SLD, Reflect-DiT, or ReasonEdit. Applying GRPO with a sequential VLM-based reward is a standard alignment technique. While the authors demonstrate performance on images generated by various base models, they fix the editing module to Qwen-Image-Edit. This experimental design leaves it unclear whether the policy learned generalized visual rectification logic or merely overfit to the specific prompting priors of the chosen editor.

---

> ### Author Rebuttal · Authors · 2026-03-31
>
> Thank you for the constructive feedback. All suggestions will be incorporated.
>
> > **Response to W1 & Q1**
>
> **Credit Assignment in GRPO.** We agree that Stage II does not fully disentangle policy failure from editor failure. However, updates are based on group-relative advantages:
>
> $r_i=r_i^{\\mathrm{reason}}+\\beta r_i^{\\mathrm{rect}},\\qquad A_i=\\frac{r_i-\\mu_x}{\\sigma_x+\\varepsilon}.$
>
> For a fixed prompt, variation across rollouts is the intended supervision signal: instructions that lead to better edits receive a higher reward. If the editor fails uniformly, the rectification reward becomes a prompt-level constant that is removed by GRPO normalization.
>
> **Cross-Editor Generalization.** We train two variants with different editors (Qwen-Image-Edit and BAGEL) and evaluate them across multiple inference editors on GenEval++ (Tab. R1) and R3-Bench (Tab. R2). All settings improve, indicating cross-editor transfer.
>
> **Tab. R1.** Cross-editor generalization on GenEval++ (Avg, $N=2$). GPT Image and OmniGen2 use native editing at inference. Qwen-Image uses Qwen-Edit. R3-Refiner-QE and R3-Refiner-BG denote variants trained with Qwen-Edit and BAGEL, respectively.
>
> | Method | GPT Image | Qwen-Image | OmniGen2 |
> |---|---:|---:|---:|
> | Baseline | 0.793 | 0.654 | 0.314 |
> | + R3-Refiner-QE | 0.829 | 0.714 | 0.364 |
> | + R3-Refiner-BG | 0.839 | 0.711 | 0.343 |
>
> **Tab. R2.** Cross-editor evaluation on R3-Bench ($S_\\text{rect}$). Deltas are relative to Qwen3-VL-8B.
>
> | Method | Qwen-Edit | Banana | GPT Image |
> |---|---:|---:|---:|
> | GPT-5.2 | 0.65 | 0.55 | 0.72 |
> | Gemini-3-Pro | 0.60 | 0.56 | 0.71 |
> | Qwen3-VL-8B | 0.54 | 0.49 | 0.63 |
> | R3-Refiner-QE | 0.62 (+0.08) | 0.58 (+0.09) | 0.72 (+0.09) |
> | R3-Refiner-BG | 0.66 (+0.12) | 0.58 (+0.09) | 0.71 (+0.08) |
>
> > **Response to W2 & Q2**
>
> We address this with two analyses.
>
> **Bootstrap Confidence Intervals (CI).** Using paired bootstrap resampling [1], we resample the 670 instances with replacement ($B=1000$), share indices across models, and compute 95% CIs for pairwise $\\Delta S_\\text{rect}$. As shown in Tab. R3, R3-Refiner is statistically distinguishable from Gemini-3-Pro, Qwen3-VL-8B, and OmniVerifier at $\\alpha=0.05$, while its comparison with GPT-5.2 is non-significant. Given their $S_\\text{rect}$ gap of only 0.005, this is more consistent with a near-tie than with insufficient benchmark sensitivity.
>
> **Rank Stability.** We further draw stratified subsamples (500 rounds per size) and compute Kendall's $\\tau$ against the full-set ranking. As shown in Tab. R4, $\\tau$ reaches 0.95 at $n=400$. The only instability comes from the R3-Refiner/GPT-5.2 pair, which is expected given their 0.005 gap and non-significant difference in Tab. R3. Our benchmark size (670) is comparable to GenEval (553) and Step1X-Edit (606).
>
> **Tab. R3.** Paired bootstrap significance test for R3-Refiner-BG ($S_\\text{rect}=0.658$) vs. other models ($B=1000$, 95% CI of $\\Delta S_\\text{rect}$). Sig.: CI excludes zero.
>
> | Other Models | $S_\\text{rect}$ | 95% CI ($\\Delta$) | Sig. |
> |---|---:|---:|---:|
> | GPT-5.2 | 0.653 | [−0.040, +0.050] | ✗ |
> | Gemini-3-Pro | 0.599 | [+0.008, +0.109] | ✓ |
> | Qwen3-VL-8B | 0.544 | [+0.064, +0.163] | ✓ |
> | OmniVerifier | 0.167 | [+0.427, +0.555] | ✓ |
>
> **Tab. R4.** Rank stability under subsampling. $\\dagger$: treating the non-significant R3-Refiner/GPT-5.2 pair as tied.
>
> | Metric | n=200 | n=400 | Full |
> |---|---:|---:|---:|
> | Kendall $\\tau$ | 0.92 | 0.95 | 1.00 |
> | Kendall $\\tau^\\dagger$ | 0.97 | 1.00 | 1.00 |
> | Exact Match$^\\dagger$ | 87% | 100% | 100% |
>
> > **Response to W3 & Q3**
>
> **Distinction from Prior Work.** Our contribution is not the loop itself, but the insights behind it (Tab. R5). R3-Bench shows that current models diagnose visual errors well but often fail to correct them. Naive RL reward design can then lead to Illusory Visual Rectification (IVR), where the policy rewrites the prompt to match the flawed image. R3-Refiner addresses this with a hierarchical reward grounded in measurable visual improvement.
>
> **Tab. R5.**
>
> | Key Contribution | SLD | Reflect-DiT | ReasonEdit | Ours |
> |---|---|---|---|---|
> | Reflective Visual Generation (RVG) | ✓ | ✓ | ✓ | ✓ |
> | Decoupled Stage-Wise Evaluation of RVG | ✗ | ✗ | ✗ | ✓ (R3-Bench) |
> | Identified Capability Misalignment | ✗ | ✗ | ✗ | ✓ (Tab. 1) |
> | Identified and Resolved IVR | ✗ | ✗ | ✗ | ✓ (Fig. 5) |
> | Cross-Editor Generalization | ✓ | ✗ | ✗ | ✓ (Tabs. 2, 3, R1, R2) |
>
> > **Response to Q4**
>
> We audited 300 samples (50 retained and 50 rejected per source), each labeled by 9 annotators via majority vote. Tab. R6 reports the false-positive rate (FPR) and false-negative rate (FNR). The low FPR (2–8%) indicates high precision, while the higher FNR reflects conservative filtering.
>
> **Tab. R6.**
>
> | Source | FPR | FNR |
> |---|---:|---:|
> | T2I-R1 | 4.0% | 42.0% |
> | BLIP-3O | 2.0% | 46.0% |
> | Pico-Banana | 8.0% | 16.0% |
>
> [1] *Statistical Significance Tests for Machine Translation Evaluation*, EMNLP 2004.

---

> > ### Author Rebuttal · Reviewer_bbzb · 2026-04-02
> >
> > Thank you for the detailed rebuttal and the additional experiments provided. I appreciate the effort to address the core methodological concerns raised in my review. However, fundamental issues regarding RL credit assignment and benchmark scale remain unresolved and require revisions beyond what can be addressed in a rebuttal.
> >
> > 1. **RL Credit Assignment (W1/Q1):** The authors argue that Group Relative Policy Optimization (GRPO) normalization removes the editor's failure as a prompt-level constant because the editor fails uniformly across rollouts. This assumption is flawed. In generative models (especially diffusion-based editors), failure is highly stochastic and heavily dependent on the specific phrasing of the editing instruction. An editor will not fail uniformly across N rollouts if the policy explores different text instructions. Thus, the variance in the reward signal ($S_{rect}$) remains permanently confounded by the editor's stochastic success rate, meaning the policy is still optimizing against the editor's noise rather than purely learning generalized rectification logic. The cross-editor generalization results in Tables R1 and R2 show that the policy can transfer, but they do not resolve the mathematical confounding present during the training phase itself.
> >
> > 2. **Benchmark Scale (W2/Q2):** The authors provide bootstrap confidence intervals and rank stability metrics to defend the 670-sample size of R3-Bench. While the statistical tests are executed correctly, they miss the broader point regarding domain coverage. A sample size of 670 divided across 8 fine-grained compositional categories leaves approximately 83 samples per category. Bootstrapping a sample size of 83 does not magically generate the diversity required to evaluate open-ended spatial, numerical, and attribute-binding capabilities. The fact that the benchmark size is "comparable to GenEval (553)" is not a strong defense, as the community has increasingly recognized that these early benchmarks are statistically underpowered for evaluating modern frontier models. A robust evaluation of compositional generation requires thousands of prompts to prevent overfitting to a narrow set of templates.
> >
> > 3. **Originality (W3/Q3):** The authors provide Table R5 to highlight their contributions (e.g., identifying Illusory Visual Rectification). While identifying IVR is a valid empirical observation of reward hacking, solving it by anchoring the reward to the original prompt is a standard RLHF correction, not a novel algorithmic breakthrough. The core Reason-Reflect-Rectify pipeline remains structurally identical to existing critique-and-revise methods.
> >
> > I acknowledge the authors' rigorous analysis of the false-positive/false-negative rates for their filtering pipeline (Q4), which was well executed. However, because the core RL objective remains mathematically confounded by the frozen editor's variance, and the evaluation benchmark remains too small to definitively prove open-ended compositional mastery, I maintain my score. The paper requires a significant structural update to the RL training methodology to decouple policy reasoning from editor execution.

---

> > > ### Author Response · Authors · 2026-04-03
> > >
> > > We appreciate the reviewer's comments and clarify the following:
> > >
> > > > **Response to W1 & Q1**
> > >
> > > The remaining disagreement concerns formal disentanglement during training, whereas **the central question for this paper is whether R3-Refiner effectively improves Reflective Visual Generation (RVG)**.
> > >
> > > **On RL Credit Assignment.** We acknowledge that the frozen editor's execution quality varies with instruction phrasing. However, this variation provides a meaningful learning signal: precise and actionable instructions are more likely to yield better editor outputs.
> > >
> > > **Cross-Editor Transfer.** Cross-editor experiments on two R3-Refiner variants across five editors and two benchmarks show consistent improvement (Tabs. 1, 2, 3, R1, R2). Specifically, R3-Refiner-BG, trained with BAGEL, also improves with Qwen-Image-Edit at inference.
> > >
> > > **Empirical Evidence of Learned Policy.** We compare R3-Refiner against other iterative strategies (Tab. R7). Best-of-N saturates and declines with more rounds. Prompt resubmission (re-feeding the original prompt and edited image to the editor) improves slightly at first but drops back by round two. The pretrained Qwen3-VL-8B degrades with iteration due to false positives in verification. **Only R3-Refiner variants improve consistently across rounds**, supporting that gains stem from the learned policy.
> > >
> > > Tab. R7. Iterative ablation on GenEval++. All use Qwen-Image-Edit as the editor. R3-Refiner-QE and R3-Refiner-BG denote variants trained with Qwen-Edit and BAGEL.
> > >
> > > | Model | N | Avg |
> > > |:---|:---:|:---:|
> > > | Qwen-Image | 0 | 0.654 |
> > > | + Best-of-$N$ | 3, 6 | 0.682, 0.679 |
> > > | + Prompt resub. | 1, 2 | 0.668, 0.654 |
> > > | + Qwen3-VL-8B | 1, 2 | 0.639, 0.571 |
> > > | + R3-Refiner-QE | 1, 2 | 0.704, 0.714 |
> > > | + R3-Refiner-BG | 1, 2 | 0.686, 0.711 |
> > >
> > > > **Response to W2 & Q2**
> > >
> > > R3-Bench targets the evaluation of RVG and relative model comparison on compositional rectification, not exhaustive coverage of open-ended compositional diversity. **The relevant question is whether its evaluation suffices for the comparisons made in this paper.**
> > >
> > > **Benchmark Scope and Reliability.** R3-Bench features expert-annotated instances across 8 categories sourced from T2I-R1, GEdit, and GenEval++, covering both real and generated images (Appendix G). This scale aligns with established compositional editing benchmarks (GenEval 553, RISEBench 360, Step1X-Edit 606, ImgEdit 734).
> > >
> > > **Evaluator Robustness and Human Alignment.** Substituting the evaluator with GPT-5.2 yields consistent $S_\text{rect}$ rankings (Tab. R8). Furthermore, a 23-annotator human study shows high correlation between $S_\text{rect}$ and human judgments (Tab. R9), confirming the benchmark's reliability.
> > >
> > > **Statistical Robustness.** Bootstrap CIs and ranking stability analysis (Tabs. R3, R4) further support that the current scale is sufficient for the comparisons made in this paper.
> > >
> > > Tab. R8. $S_\text{rect}$ under two evaluators.
> > >
> > > | Model | GPT $S_\text{rect}$ | GPT Rank | Qwen $S_\text{rect}$ | Qwen Rank |
> > > |:---|:---:|:---:|:---:|:---:|
> > > | R3-Refiner-BG | 0.62 | 1 | 0.66 | 1 |
> > > | GPT-5.2 | 0.62 | 1 | 0.65 | 2 |
> > > | R3-Refiner-QE | 0.58 | 3 | 0.62 | 3 |
> > > | Gemini-3-Pro | 0.55 | 4 | 0.60 | 4 |
> > > | Qwen3-VL-8B | 0.50 | 5 | 0.54 | 5 |
> > > | Qwen2.5-VL-7B | 0.36 | 6 | 0.38 | 6 |
> > >
> > > Tab. R9. $S_\text{rect}$ vs. human QA accuracy.
> > >
> > > | Evaluator | GPT-5.2 | R3-Refiner-BG | Gemini-3-Pro | Qwen3-VL-8B |
> > > |:---|:---:|:---:|:---:|:---:|
> > > | $S_\text{rect}$ | 0.650 | 0.657 | 0.599 | 0.543 |
> > > | Human | 0.912 | 0.912 | 0.829 | 0.719 |
> > >
> > > *(SROCC=0.800, KROCC=0.667, Fleiss' $\kappa$=0.776).*
> > >
> > > > **Response to W3 & Q3**
> > >
> > > **Distinction from Prior Work.** Our contribution **is not the loop itself but the analysis and evaluation framework for RVG.** Specifically, R3-Bench decomposes RVG into independently evaluable stages. This reveals the misalignment between diagnosis and rectification (Tab. 1) and identifies Illusory Visual Rectification (IVR) as a concrete failure mode of naive RL reward design (Fig. 5). To address IVR, we design a hierarchical reward in which Stage I supervises diagnosis against ground-truth labels, while Stage II uses the model's diagnostic capability to reward rectification learning. This design is motivated by the observed capability gap and produces robust gains across editors and benchmarks.
> > >
> > > **Limits of Prior Work.** OmniGen2 [3] attempts reflective generation via SFT but obtains $S_\text{rect} = -0.19$ on R3-Bench (Tab. 1) and leaves RL as future work (Sec. 5.5). Similarly, Reflect-DiT [2] explores RL (D3PO, Diffusion-DPO) but abandons it due to training instability (Appendix A.4). **Neither work proposes a systematic evaluation of RVG, identifies the capability misalignment, or addresses IVR.**
> > >
> > > We hope these clarifications are helpful for the final assessment.
> > >
> > > *[2] Reflect-DiT: Inference-Time Scaling for T2I via In-Context Reflection. ICCV 2025*
> > >
> > > *[3] OmniGen2: Exploration to Advanced Multimodal Generation. arXiv 2025*

---

### Official Review · Reviewer_dZbT · 2026-03-11

**Soundness:** 3
**Presentation:** 3
**Significance:** 3
**Originality:** 3
**Overall Recommendation:** 5
**Confidence:** 4

**Summary:**

This paper studies how image generation models are able to produce high quality images and identify generation errors but are unable to generate rectification examples. To address this issue the paper introduced R3-Bench, a benchmark designed to evaluate whether models can identify errors and propose corrections. They also propose R3-Refiner, a training framework that encourages models to generate actionable fixes rather than only diagnosing problems. The show that their method can be applied to any model and improves performance on both R3-bench as well as other benchmarks.

**Compliance With Llm Reviewing Policy:**

Affirmed.

**Final Justification:**

I had a positive assessment of the paper in my initial review, and the rebuttal further strengthened my confidence in the work.The paper introduces a clear benchmark targeting the gap between identifying errors and producing rectification suggestions, which I view as a meaningful and underexplored capability of generative models. The proposed training framework is simple, practical, and shows consistent improvements across multiple benchmarks.

The rebuttal addressed my main concerns and clarified several implementation details, reinforcing my confidence in the soundness and generality of the approach. Overall, I continue to view the contribution as original and useful, and I am maintaining my original score of 5.

**Key Questions For Authors:**

Overall, I quite like this paper. A few questions:
1. How much of the improvement comes from iterative inference versus the learned policy?
A comparison against baselines that use similar multi-step inference would be appreciated.
2. How robust are the reported results to the choice of automated evaluators?
A human study comparing a baseline model with and without R3 could provide more confidence for the method.

**Limitations:**

yes

**Strengths And Weaknesses:**

Strengths:
The benchmark measures a new capability of generative models, specifically the gap between identifying errors and producing rectification suggestions.
The training framework is simple and practical, and the authors show that it can be applied to multiple models and improves performance across several benchmarks.

Weaknesses:
It is unclear whether the gains come from improved reasoning or simply from additional iterative inference. A stronger baseline could involve re-submitting the edited image together with the original instruction to the editing model.
All reported metrics are automated and no human evaluation is provided. An Elo-style study comparing a baseline model with and without R3 would provide additional confidence in the results.

---

> ### Author Rebuttal · Authors · 2026-03-31
>
> Thank you for your constructive feedback. All your suggestions will be incorporated.
>
> > **Response to W1 & Q1**
>
> **Iterative Inference vs. Learned Policy.** Tab. 5 in our submission compares two inference-time scaling strategies on GenEval++: R3-Refiner with serial iterative refinement ($N=1$--$3$) and Best-of-$N$ with parallel sampling ($N=3$--$6$). R3-Refiner already exceeds the best Best-of-$N$ result with a single refinement round. To further isolate the contribution of the RL-trained policy from iterative inference itself, we conduct additional ablations (Tab. R1). All configurations use the same editor (Qwen-Image-Edit) and differ only in the verification strategy. Beyond Best-of-$N$, we add prompt resubmission (re-feeding the original prompt to the editor without verification, as suggested by the reviewer) and a pretrained Qwen3-VL-8B verifier, together with two R3-Refiner variants trained with different editors (QE and BG).
>
> **Results.** As shown in Tab. R1, Best-of-$N$ saturates and slightly declines as $N$ increases. Prompt resubmission gives a small gain at $N=1$ but returns to baseline at $N=2$, suggesting that repeated editing without verification can corrupt already aligned content. The pretrained Qwen3-VL-8B verifier degrades further at $N=2$, consistent with excessive false positives. In contrast, only R3-Refiner variants improve across rounds, confirming that gains stem from the learned policy.
>
> **Tab. R1.** Iterative ablation on GenEval++ (Avg). All configurations use Qwen-Image-Edit as the editor. R3-Refiner-QE and R3-Refiner-BG denote variants trained with Qwen-Image-Edit and BAGEL, respectively. Some categories are omitted for brevity.
>
> | Model | N | Color | Count | ... | Multi | Avg |
> |---|---:|---:|---:|---:|---:|---:|
> | Qwen-Image-2512 | 0 | 0.800 | 0.700 | ... | 0.550 | 0.654 |
> | + Best-of-N | 3 | 0.875 | 0.700 | ... | 0.500 | 0.682 |
> |  | 5 | 0.850 | 0.725 | ... | 0.550 | 0.686 |
> |  | 6 | 0.825 | 0.725 | ... | 0.550 | 0.679 |
> | + Prompt resub. | 1 | 0.825 | 0.750 | ... | 0.600 | 0.668 |
> |  | 2 | 0.775 | 0.725 | ... | 0.575 | 0.654 |
> | + Qwen3-VL-8B | 1 | 0.775 | 0.675 | ... | 0.575 | 0.639 |
> |  | 2 | 0.675 | 0.650 | ... | 0.525 | 0.571 |
> | + R3-Refiner-QE | 1 | 0.925 | 0.800 | ... | 0.550 | 0.704 |
> |  | 2 | 0.925 | 0.775 | ... | 0.550 | 0.714 |
> | + R3-Refiner-BG | 1 | 0.825 | 0.725 | ... | 0.675 | 0.686 |
> |  | 2 | 0.850 | 0.750 | ... | 0.675 | 0.711 |
>
> > **Response to Q2**
>
> **Evaluator Robustness.** To assess robustness to the evaluator, we focus on the Rectification Score ($S_\\text{rect}$), which is the key metric in R3-Bench for measuring relative visual improvement. We replace Qwen3-VL-235B with GPT-5.2 (thinking effort set to low) as the evaluator and re-run all evaluations without changing any other component of the pipeline. As shown in Tab. R2, both evaluators produce highly consistent $S_\\text{rect}$ rankings across all 9 models.
>
> **Human Evaluation.** We conduct a human study on 24 (category-balanced) sampled R3-Bench instances using 23 annotators. We select four representative models spanning a broad quality range: the base model (Qwen3-VL-8B), our RL-trained R3-Refiner-BG, and two strong closed-source models (GPT-5.2 and Gemini-3-Pro). Annotators answer factual yes/no questions derived from the original prompt. We compute Spearman (SROCC) and Kendall (KROCC) rank correlation coefficients between $S_\\text{rect}$ and human QA accuracy, and measure inter-annotator agreement using Fleiss' kappa [1]. As shown in Tab. R3, $S_\\text{rect}$ and human judgments are closely aligned, with consistent model rankings. Fleiss' $\\kappa$ further indicates substantial inter-annotator agreement under the interpretation of Landis and Koch [2].
>
> **Tab. R2.** $S_\\text{rect}$ rankings under two evaluators across 9 representative models. Tied scores receive the same rank.
>
> | Model | GPT $S_\\text{rect}$ | Qwen $S_\\text{rect}$ | GPT Rank | Qwen Rank |
> |---|---:|---:|---:|---:|
> | R3-Refiner-BG | 0.62 | 0.66 | 1 | 1 |
> | GPT-5.2 | 0.62 | 0.65 | 1 | 2 |
> | R3-Refiner-QE | 0.58 | 0.62 | 3 | 3 |
> | Gemini-3-Pro | 0.55 | 0.60 | 4 | 4 |
> | Qwen3-VL-8B | 0.50 | 0.54 | 5 | 5 |
> | GPT-4o | 0.49 | 0.53 | 6 | 6 |
> | Qwen2.5-VL-7B | 0.36 | 0.38 | 7 | 7 |
> | SLD | 0.25 | 0.28 | 8 | 8 |
> | OmniVerifier | 0.17 | 0.17 | 9 | 9 |
>
> **Tab. R3.** Alignment between automated $S_\\text{rect}$ and human QA accuracy across 4 representative models.
>
> | Evaluator | GPT-5.2 | R3-Refiner-BG | Gemini-3-Pro | Qwen3-VL-8B | SROCC | KROCC | Fleiss' $\\kappa$ |
> |---|---:|---:|---:|---:|---:|---:|---:|
> | $S_\\text{rect}$ | 0.650 | 0.657 | 0.599 | 0.543 | - | - | - |
> | Human | 0.912 | 0.912 | 0.829 | 0.719 | 0.800 | 0.667 | 0.776 |
>
> [1] Fleiss, J. L. (1971). *Measuring nominal scale agreement among many raters*. Psychological Bulletin, 76, 378--382.
> [2] Landis, J. R., and Koch, G. G. (1977). *The measurement of observer agreement for categorical data*. Biometrics, 33, 159--174.

---

> > ### Author Rebuttal · Reviewer_dZbT · 2026-04-02
> >
> > I have no follow up questions. I will maintain my score.

---

> > > ### Author Response · Authors · 2026-04-03
> > >
> > > Thank you for the positive review and for confirming that your questions have been fully resolved. We appreciate your recognition that R3-Bench measures a new capability and that our framework is simple and practical. The additional baselines and human evaluation you suggested have strengthened the paper. We will include all these new results in the final manuscript. Thank you again for your time and thoughtful feedback.

---

### Decision · Program_Chairs · 2026-04-30

**Decision:**

Accept (regular)

**Comment:**

In this paper, authors formalized the reason-reflect-rectify loop as a core framework and introduce R^3-Bench to enable multi-round reflective visual generation. They also proposed R^3-Refiner to better align rectification with reflective reasoning. Experimental results showed the effectiveness of the proposed method.

The final rating of this paper are: 1 accept, 2 weak accept, 1 weak reject.

Before rebuttal, reviewers thought

the strength of this paper are:
1) benchmark measures a new capability of generative models. (Reviewer dZbT)
2) The training framework is simple and practical. (Reviewer dZbT)
3) well motivated. (Reviewer bbzb, Zc4f)
4) benchmark provides a specific way to evaluate the entire iterative Reason-Reflect-Rectify process. (Reviewer Zc4f)
5) proposed R^3-Refiner works effectively as a plug-and-play module. (Reviewer Zc4f)
6) paper is well-written and easy to follow. (Reviewer eRBv)
7) R3-Bench is curated with human expert reviewing. (Reviewer eRBv)
8) proposed R3-refiner is shown to be effective. (Reviewer eRBv)

weaknesses are:
1) unclear whether the gains come from improved reasoning or simply from additional iterative inference. (Reviewer dZbT)
2) All reported metrics are automated and no human evaluation is provided. (Reviewer dZbT)
3) proposed evaluation benchmark is severely underpowered. (Reviewer bbzb)
4) The originality of the submission is low. (Reviewer bbzb)
5) benchmark size is relatively small. (Reviewer Zc4f)
6) framework relies heavily on a frozen external image editor. (Reviewer Zc4f)
7) The overall visual improvement is strictly bounded by the inherent capabilities of the chosen external editor. (Reviewer Zc4f)
8) R3-loop has been extensively discussed in prior works. (Reviewer eRBv)
9) concerns on specific editor model. (Reviewer eRBv)

After rebuttal,
Reviewer dZbT thought their concerns are fully addressed and maintained accept rating.

Reviewer bbzb mentioned their concerns are partially addressed, i.e., fundamental issues regarding RL credit assignment and benchmark scale remain unresolved. They maintained weak reject score.

Reviewer Zc4f said their concerns are fully addressed and raised score to weak accept.

Reviewer eRBv mentioned their concerns are fully addressed. But in the meantime also mention framing R^3 as major contribution. Also concerns on the limited scale of the benchmark. And they raised score the weak accept.

Considering all these, AC make weak accept decision and hope authors could incorporate reviewers' comments and improve the paper.